Manuscript prepared for Atmos. Chem. Phys.

with version 2014/05/15 6.81 Copernicus papers of the LaTeX class copernicus.cls.

Date: 8 February 2017

# Observed versus simulated mountain waves over Scandinavia - improvement of vertical winds, energy and momentum fluxes by enhanced model resolution?

Johannes Wagner[1], Andreas Dörnbrack[1], Markus Rapp[1], Sonja Gisinger[1], Benedikt Ehard[1], Martina Bramberger[1], Benjamin Witschas[1], Fernando Chouza[1], Stephan Rahm[1], Christian Mallaun[2], Gerd Baumgarten[3], and Peter Hoor[4]

[1]Deutsches Zentrum für Luft- und Raumfahrt, Institut für Physik der Atmosphäre, 82234 Oberpfaffenhofen, Germany

[2]Deutsches Zentrum für Luft- und Raumfahrt, Flugexperimente, 82234 Oberpfaffenhofen, Germany

[3]Leibniz Institut für Atmosphären Physik, 18225 Kühlungsborn, Germany

[4]Johannes Gutenberg Universität, Institut für Physik der Atmosphäre, 55099 Mainz, Germany

*Correspondence to:* Johannes Wagner (johannes.wagner@dlr.de)

**Abstract.** Two mountain wave events, which occurred over northern Scandinavia in December 2013 are analysed by means of airborne observations and global and mesoscale numerical simulations with horizontal mesh sizes of 16 km, 7.2 km, 2.4 km and 0.8 km. During both events westerly cross-mountain flow induced upward propagating mountain waves with different wave characteristics due to differing atmospheric background conditions. While wave breaking occured at altitudes between 25 km to 30 km during the first event due to weak stratospheric winds, waves propagated to altitudes above 30 km and interfacial waves formed in the troposphere at a stratospheric intrusion layer during the second event. Global and mesoscale simulations with

16 km and 7.2 km grid sizes were not able to simulate the amplitudes and wavelengths of the mountain waves correctly due to unresolved mountain peaks. In simulations with 2.4 km and 0.8 km horizontal resolution mountain waves with horizontal wavelengths larger than 15 km were resolved, but exhibited too small amplitudes and too high energy and momentum fluxes. Simulated fluxes could be reduced by either increasing the vertical model grid resolution or by enhancing turbulent diffusion in the model, which is comparable to an improved representation of small-scale nonlinear wave effects.

## 1   Introduction

Internal gravity waves (GWs) exchange energy and momentum between the troposphere and the middle atmosphere (Fritts and Alexander, 2003). Especially the interaction of GWs with the mean flow plays an important role in atmospheric dynamics, as dissipating GWs deposit momentum, e.g., by wave breaking, which changes the background flow (Eliassen and Palm, 1960). This momentum deposition drives the meridional Brewer Dobson circulation (Holton and Alexander, 2000) or the periodically changing westerly and easterly winds in the tropic stratosphere, known as the quasi-biennial oscillation (QBO, e.g., Baldwin et al., 2001).

Due to their importance for atmospheric flows from the boundary layer to the middle atmosphere, GWs have been studied intensively in the past by means of analytical and numerical models (e.g., Queney, 1948; Scorer, 1949; Durran, 1990) and a large number of field campaigns like the Momentum Budget over the Pyrénées experiment (PYREX, Bougeault et al., 1990, 1993), the Mesoscale Alpine Programme (MAP, Bougeault et al., 2001), the Terrain-induced Rotor EXperiment (T-REX, Grubišić et al., 2008) or the DEEPWAVE campaign (Fritts et al., 2016). An overview of some previous GW field campaigns is given in Smith et al. (2016). In December 2013 the Gravity Wave Life Cycle I (GW-LCYCLE I) campaign took place over northern Scandinavia to observe the whole life cycle of GWs from their excitation via propagation to dissipation. The Scandinavian coastal mountain range together with the wintertime synoptic situation over northern Scandinavia and the proximity to the polar vortex represent favourable conditions for the generation and propagation of mountain waves. In the stratosphere, these waves can form ice clouds (e.g., Dörnbrack et al., 2001; Dörnbrack and Leutbecher, 2001; Dörnbrack et al., 2002) and regions of gravity wave breaking as observed by Ehard et al. (2016) at altitudes of 30 km. The principal idea of the GW-LCYCLE I campaign was to conduct observations whenever the meteorological conditions favoured mountain wave excitation and vertical wave propagation. In this kind, GW-LCYCLE I was a forerunner experiment to test flight strategies and synergies between the different instruments, which were afterwards applied during the two consecutive gravity wave campaigns

DEEPWAVE in New Zealand during austral winter 2014 (Fritts et al., 2016) and GW-LCYCLE II in Scandinavia during winter 2015/2016.

Despite the large number of field campaigns and modelling studies, the accurate simulation of propagating and breaking GWs is still a challenge for weather and climate models. Simulations with high model grid resolutions can help to understand the complex interaction of different kinds of waves on a multitude of scales during a GW event by a detailed analysis of the respective meteorological situation. These simulations are, however, dependent on initial and boundary conditions of larger-scale models, whose GW parameterization schemes have to be improved (Kim et al., 2003). Vosper et al. (2016) investigated the parameterization of GW-induced pressure drag and momentum fluxes in dependence of the horizontal model grid resolution $\Delta x$. They showed a discrepancy between resolved and parameterized drag processes, as parameterization schemes typically only represent processes due to subgrid-scale orography. In addition, they should also include processes from longer wavelengths of up to $10\Delta x$, which contribute to GW-induced drag but cannot be resolved by numerical models. This issue is important for simulations with model grid resolutions in the order of tens of kilometres where the topography and related GW processes are only partly resolved. By tuning their parameterization scheme Vosper et al. (2016) were able to improve the GW-induced pressure drag, but not the mountain wave momentum fluxes.

GW-induced momentum and energy fluxes have been investigated in several studies by means of observations (e.g., Smith et al., 2008, 2016) and simulations (e.g. Kruse and Smith, 2015). A comparison of simulated and observed momentum fluxes is presented in Kruse and Smith (2016) for the DEEPWAVE campaign. On average, WRF simulations with 6 km horizontal resolution resulted in 5% weaker momentum fluxes compared to observations, while a reduction of the horizontal resolution to 2 km lead to 50% stronger momentum fluxes. Along single flight legs the deviation of simulated fluxes from observed fluxes was higher with up to 60% and 85% for the 6 km and 2 km run, which shows according to (Kruse and Smith, 2016) that momentum fluxes are not predictable in a complex 3D wave field even for model resolutions of up to 2 km. Therefore, we investigate if simulated fluxes can be improved by model grid resolutions in the sub-kilometre range during the GW-LCYCLE campaign. In addition, this enables to study systematically the influence of model resolution and model topography on simulated GWs in a similar way as presented in Udina et al. (2017) for simulations of trapped waves in the lee of the Pyrénées. In contrast to Udina et al. (2017), who used surface stations and vertical profiles from a wind profiler and two radiosondes to verify their model results, our simulations are compared to 2D lidar and in-situ measurements on flight legs across the Scandinavian mountain range.

The overall goal of this paper is to test the ability of a state-of-the-art mesoscale model to simulate three-dimensional GW structures in the upper troposphere and lower stratosphere (UTLS) region. Especially, we investigate the impact of horizontal model grid and topography resolutions on the simulation results. Of certain interest is the accurate simulation of GW-induced energy (EF) and momentum fluxes (MF). For this purpose the horizontal model grid resolution is increased to 800 m to resolve single mountain peaks and to investigate, if first a sub-kilometre horizontal resolution significantly improves the representation of waves. Secondly, whether it is possible to compute vertical winds, whose magnitude and spatial structure are comparable to lidar and in-situ observations. Sensitivity runs with increased vertical grid distances and increased turbulent diffusion are performed to study possible impacts of unresolved, non-linear processes on energy and momentum fluxes. Simulation results are compared to vertical sections of airborne DWL measurements in the tropopause and to in-situ observations at flight level.

The paper is organized as follows: Section 2 gives an overview of the GW-LCYCLE I campaign and the available data sets. In section 3 the numerical simulations and methods, which are used to analyse GWs are presented. The general synoptic situation during the campaign is described in section 4 by means of numerical simulations. Observed and simulated GW structures in the UTLS are compared and analysed in section 5 and the conclusion is presented in section 6.

## 2   Campaign and data set overview

### 2.1   GW-LCYCLE I campaign

The GW-LCYCLE I campaign took place from 2 to 14 December 2013 in Kiruna, northern Sweden (68° N, 20° E; see Fig. 1). The principal observational platform was the Deutsches Zentrum für Luft- und Raumfahrt (DLR) research aircraft Falcon, based at Kiruna airport in the lee of the Scandinavian mountain range. Airborne observations were complemented by radiosonde measurements launched at the windward side of the mountains at Andenes (69° N, 16° E, Norway), at the leeward side at the European Space and Sounding Rocket Range Esrange (68° N, 21° E, Sweden) at Kiruna airport (Sweden) and further downstream at Sodankylä (67° N, 27° E, Finland). In addition, ground-based lidar systems were operated at the Arctic Lidar Observatory for Middle Atmosphere Research (ALOMAR) in Andenes and at Esrange and provided time series of temperature and wind profiles at altitudes between 30 km to 90 km. At Andenes, middle atmospheric winds were measured with the Middle Atmosphere Alomar Radar System (MAARSY). The flight strategy during the campaign focused on synoptic situations with strong westerly cross mountain flow, which are favourable for the excitation of mountain waves (Dörnbrack et al., 2001, 2002). The planning of the respective research flights was facilitated by the usage of the Mission Support System

(MSS, Rautenhaus et al., 2012), which is a software tool to compute meteorological parameters along virtual flight legs on the basis of numerical weather prediction model output.

Altogether, there were 5 intensive observation periods (IOP) with a total of 6 research flights and 92 radiosoundings (see Table 1). IOP 1 and 5 were mountain wave events which were studied with 4 research flights and are investigated in this paper. No research flight could be conducted during a strong mountain wave event on 11 December 2013 (IOP4) as a downslope wind storm with gale force cross winds made take-off and landing impossible at Kiruna airport. The event during IOP2 was dominated by the winter storm "Xaver", which passed over northern Germany (Fenoglio-Marc et al., 2015). This storm caused mountain and jet-induced GWs over southern Scandinavia and deep propagating convective GWs in a strong, convective cold air outbreak with polar low formation over the Norwegian Sea. Finally, relatively calm conditions prevailed during IOP3, enabling the measurement of polluted air (mainly of $SO_2$), which was advected towards northern Scandinavia from midlatitudes with sources in the US and China (H. Schlager, 2013, personal communication).

## 2.2 Airborne observations

The DLR Falcon aircraft was equipped with a downward-looking coherent Doppler wind lidar (DWL) which operates at a wavelength of 2 $\mu$m. Within the last years, this lidar system was successfully deployed in several ground-based and airborne field campaigns for instance for measuring aircraft wake vortices (Köpp et al., 2003), aerosol optical properties (Chouza et al., 2015) and the three-dimensional wind field over the Atlantic ocean (Weissmann et al., 2005). Details about the DWL hardware configuration are given by Chouza et al. (2015) and Witschas et al. (2017) and details about the retrieval procedure can be found in Smalikho (2003).

During the GW-LCYCLE I campaign, the DWL was operated in scanning or nadir modes aiming to measure the vertical profiles of the three-dimensional wind vector or to measure the vertical wind speed, respectively. While operating in scanning mode, a conical step-and-stare scan around the vertical axes with an off-nadir angle of $20°$ is performed, which results in a horizontal resolution of 9 km. During nadir mode operation, the laser beam is pointed to nadir-direction and the measured LOS wind equals the vertical wind speed with a horizontal resolution of 200 m. The vertical resolution for both measurement modes is determined by the laser pulse length and is set to be 100 m. A detailed technical description of the DWL measurements during GW-LCYCLE I is given in Witschas et al. (2017).

During the GW-LCYCLE I campaign the nadir operating mode was used most frequently, as it was suitable to detect small scale gravity waves over the complex orography. An overview of the flight altitudes and flight legs where the lidar operated in

nadir and scanning modes during IOP1 and IOP5 is given in Fig. 2. In this figure the topography height along the flight legs is

125 represented by the Advanced Spaceborne Thermal Emission and Reflection Radiometer (ASTER) data set (Schmugge et al.,

2003), which has a horizontal resolution of 30 m.

Besides wind lidar observations, in-situ measurements of standard meteorological parameters were conducted at flight level

during all research flights with a time resolution of 1 s. The measurement of in-situ 3-dimensional wind is described in Mallaun

et al. (2015) and the verification of airborne pressure measurements in Giez et al. (2017). Trace gas measurements were

130 performed at flight level with the water vapour analyser (WARAN) hygrometer (Groß et al., 2014) and a mass spectrometer for

the trace gases water vapour and $SO_2$, respectively. The University of Mainz Airborne Quantum Cascade Laser-spectrometer

(UMAQS; Müller et al., 2015) was applied to measure nitrous oxide ($N_2O$), which is virtually inert in the UTLS. The transition

from nearly constant tropospheric (327.5 ppbv $\pm$ 0.9 ppbv) to decreasing stratospheric $N_2O$ mixing ratios allows for the

determination of the chemical tropopause, which was defined by mixing ratios of 326.6 ppbv during GW-LCYCLE I. The

135 main objective of trace gas measurements was to detect GW-induced vertical mixing processes in the tropopause region.

## 2.3   Ground based observations

Airborne measurements were complemented by ground-based meteorological observations. During the five IOPs a total of 92

radiosondes was released both on the wind and leeward sides of the mountain range at Andenes (Norway), Esrange (Sweden),

Kiruna (Sweden) and Sodankylä (Finland). Depending on wind conditions the sondes drifted up to 390 km horizontally and

140 reached altitudes of more than 30 km in most cases. This dataset allows to study tropospheric GWs and their propagation

through the tropopause into the lower stratosphere. In addition to radiosonde observations, the Esrange Rayleigh lidar provided

130 hours of temperature profiles of the middle atmosphere between altitudes of 30 km to 65 km. The lidar measurements

were conducted during the period of 24 November to 14 December 2013. The analysis of the lidar and radiosonde data in

combination with mesoscale numerical modeling is described in Ehard et al. (2016).

# 3   Numerical models and analysis methods for GWs

## 3.1   Real-case simulations

Mesoscale numerical simulations of the two mountain wave events IOP1 and IOP5 are performed with the Weather Research

and Forecasting (WRF) model, version 3.7 (Skamarock et al., 2008). Up to three nested domains (D1, D2, and D3) with

horizontal resolutions of 7.2 km, 2.4 km and 0.8 km (see Fig. 1) are used. For the two coarse domains 138 terrain following levels with stretched level distances of 80 m near the surface, 160 m near the tropopause and 300 m in the upper stratosphere are used and the model top is set to 2 hPa (about 39 km altitude). For the innermost domain only 78 vertical levels are applied and the model top is set to 50 hPa (about 20 km altitude) to save computational ressources. To avoid numerical instabilities adaptive time stepping was used with a maximum time step of 15 s and a maximum Courant number of 1.2. At the model top a 7 km thick Rayleigh damping layer (Klemp et al., 2008) is applied to prevent wave reflection. Physical parameterizations contain the Rapid Radiative Transfer Model longwave scheme (Mlawer et al., 1997), the Goddard shortwave scheme (Chou and Suarez, 1994), the Mellor-Yamada-Nakanishi-Niino boundary layer scheme (Nakanishi and Niino, 2009), the Noah land surface model (Chen and Dudhia, 2001), the WRF single-moment 6-class microphysics scheme (WSM6; Hong and Lim, 2006) and the Kain-Fritsch cumulus parameterization scheme (Kain and Fritsch, 1990). The latter is switched off for the innermost domain. Horizontal diffusion (WRF parameter diff_opt) was not applied in the two innermost domains to increase GW amplitudes in vertical wind fields. The initial and boundary conditions are supplied by ECMWF (T1279 L137, cycle 40r1) operational analyses on 137 model levels with a horizontal resolution of 16 km and a temporal resolution of 6 hours. WRF and ECMWF fields are interpolated in space and time on aircraft flight tracks to compare with observational data. For this purpose, a temporal output interval of 5 minutes is used in the WRF simulations. For ECMWF a 1 hourly output interval is realized by performing short-term forecasts with the ECMWF IFS. In order to compute GW-induced energy and momentum fluxes the diagnostic filtering method of Kruse and Smith (2015) is applied to WRF output.

In addition to the WRF control simulation (CTRL), six sensitivity runs are performed (see Table 2): in the NOTOPO and OCEAN simulations the topography height is set to zero everywhere in the domain. In addition, the landuse type has properties of a water surface with a roughness length of 0.0001 m everywhere in the domain in the OCEAN runs. The NOTOPO and OCEAN simulations aim to define an atmospheric background state without mountain waves and to investigate the influence of changing surface roughnesses (transition from an ocean to a land surface) on GW excitation. In the SMTOPO simulations the two innermost domains use a smoothed topography, which is the same as in the outermost (D1) model domain. This is done to analyse the effect of unresolved topography on GW structures in a high resolution model. In the CTRLVR runs the vertical grid resolution is increased and 188 levels with constant level distances of 80 m and a model top at 50 hPa are used. Horizontal turbulent diffusion is switched on in the HVDIFF runs (WRF parameter diff_opt=1) and vertical turbulent diffusion is additionally increased by a factor of two in the H2VDIFF case by doubling the tendency terms obtained from the boundary layer scheme, which is responsible for subgrid-scale turbulent mixing in the whole atmospheric column (not just the boundary

layer). The latter three sensitivity runs are performed to improve simulated energy and momentum fluxes by damping wave amplitudes by unresolved nonlinear processes.

All WRF topographies are based on the Global 30 Arc-Second Elevation (GTOPO30) digital elevation model with a maximum horizontal resolution of about 1 km, while the ASTER data set with a horizontal resolution of 30 m is used as reference topography. Figure 3 illustrates the different representation of the Scandinavian mountain range in the different model runs by means of two example flight legs during IOP1 and IOP5. The innermost WRF domains D2 and D3 ($\Delta$x = 2.4 km and 0.8 km) resolve the individual mountain peaks very well in terms of amplitude and horizontal wavelength (note the local peak in the power spectra at about 20 km). Domain D1 and the ECMWF model do not capture the fine scale mountain peaks and represent the topography as a compact, smooth mountain ridge. The influence of topography resolution on the simulated vertical wind field is investigated in section 5.

### 3.2   Idealised simulations

To investigate the complex wave patterns and especially the occurence of trapped waves in the troposphere, which developed during IOP1 and IOP5, idealised 2D simulations (IOP1ID and IOP5ID) were performed along the two example cross sections of flight 1, leg 2 during IOP1 and IOP5 (the same as in Fig. 3). The simulations were run without moisture and were initialised with averaged upstream profiles of horizontal wind speed and potential temperature of the CTRL D3 real-case simulations at 1200 UTC on December 3 and 13, respectively. Wind speed was projected to a wind direction of 300°, which is perpendicular to the Scandinavian mountain range (Dörnbrack and Leutbecher, 2001) and represents the cross mountain flow. The model top was set to an altitude of 20 km with a damping layer thickness of 5 km. A horizontal and vertical grid resolution of 800 m and 50 m was chosen, respectively and simulations were run for 10 hours.

### 3.3   Computation of fluxes and diagnostic variables

The computation of EF and MF at flight level according to the method of Smith et al. (2008) provides information about GWs in the UTLS region and is applied for both observations and simulations in this study. The leg-averaged fluxes are computed by the formulas:

$$MF_x = \frac{\overline{\rho}}{L} \int w'u'dx, \tag{1}$$

$$MF_y = \frac{\overline{\rho}}{L} \int w'v'dx, \tag{2}$$

and

$$EF = \frac{1}{L} \int w'p'dx, \tag{3}$$

with zonal and meridional momentum fluxes $MF_x$ and $MF_y$, GW-induced perturbations of zonal wind $u'$, meridional wind $v'$, vertical wind $w'$, pressure $p'$, leg averaged density $\overline{\rho}$ and leg length $L$. Momentum and energy fluxes of linear GWs are related by the Eliassen-Palm relation (Eliassen and Palm, 1960):

$$EF = -U \cdot MF, \tag{4}$$

with

$$U \cdot MF = u \cdot MF_x + v \cdot MF_y, \tag{5}$$

with leg-averaged zonal, meridional and total horizontal wind speeds $u$, $v$ and $U$, respectively. The wind and pressure perturbations $u'$, $v'$, $w'$ and $p'$ are computed by subtracting linear regressions from full wind and pressure fields along flight legs. For pressure, a hydrostatic correction is applied in advance of computing the pressure perturbations as described in (Smith et al., 2008).

Further diagnostic variables are used in this study to describe flow and GW characteristics. The gradient Richardson number is the ratio of buoyancy to shear force and is defined as $Ri = N^2 (\frac{dU}{dz})^{-2}$ with Brunt-Vaisala frequency $N$. Typically a flow is dynamically unstable for $Ri < 0.25$.

The Scorer parameter $l = \sqrt{N^2/U^2 - \frac{1}{U}\frac{d^2U}{dz^2}}$ (Scorer, 1949) can be used to distinguish between evanescent and vertically propagating waves. Trapped lee waves occur in layers where $l$ is decreasing with height, which means that only waves with horizontal wave numbers smaller than $l$ can propagate vertically. In this study the curvature term is neglected for simplification and $l$ is computed as $l = \sqrt{N^2/U^2}$.

## 4   Meteorological conditions during IOP1 and IOP5 from simulations

### 4.1   Synoptic situation

Meteorological situations favourable for the generation of mountain waves occurred at the beginning (IOP1) and end (IOP4,
IOP5) of the GW-LCYCLE I campaign (see Table 1).

The meteorological condition during IOP1 (3 December 2013) was dominated by a strong synoptic low pressure system,
which was located over the northern Norwegian Sea and travelled eastwards from the coast of Greenland towards northern
Norway (see Fig. 4). At tropopause level, which was located at an altitude of about 5 km on the upstream side of the mountains,
a strong westerly jet moved southwards during IOP1. The cross mountain flow excited GWs and the related vertical energy
fluxes were enhanced in the middle troposphere along the whole Scandinavian mountain range with largest fluxes occuring
over the Kiruna region (Fig. 4(c)).

The GW event during IOP1 can be divided into three phases, which are marked in the time-height sections for horizontal
wind speed, gradient Richardson number, vertical energy flux and Scorer parameter at the location Abisko (68° N, 19° E; Fig. 5
(a) to (d)), which is situated in the centre of the mountain range between Andenes and Kiruna (see the dots in Fig. 4). The first
GW phase (P1) from 20 UTC on 2 December to 3 UTC on 3 December was dominated by moderate westerly cross mountain
flow at low levels (30 m s$^{-1}$ at 850 hPa) within the warm air sector of the synoptic low (not shown) and moderate vertical
energy fluxes. The second phase (P2) from 3 UTC to 6 UTC on 3 December was characterized by weaker low level winds
(15 m s$^{-1}$) and low GW activity due to calm conditions after the passage of a cold front. At upper levels the tropopause jet was
located directly over northern Scandinavia. At about 6 UTC on 3 December the third phase (P3) started when low-level winds
intensified due to the approaching low pressure system (cf. Fig. 4). During this phase the tropopause dropped to an altitude of
about 5 km upstream of the mountains and stratospheric air descended to nearly 2 km altitude in the lee of the mountains, which
is visible in a cross section of the Brunt-Vaisala frequency along leg 2 of flight 1 in Fig. 6(a). The stratospheric intrusion on the
eastern side of the mountains was also present in the NOTOPO and OCEAN simulations (not shown), which means that it was
jet-induced (cf. Shapiro and Hampel, 1987) and not generated by mountain waves. Within this tropopause fold weak interfacial
waves could develop in the CTRL D3 run (Fig. 6(a)) due to decreasing Scorer parameters (not shown). GW excitation stopped
at about 3 UTC on 4 December when the low pressure system moved further east. The two research flights of IOP1 took place
during phase P3 (see Fig. 5).

During IOP5 (13 December 2013) the situation was less complex as northern Scandinavia was located below a strong and quasi stationary northwesterly tropopause jet (Fig. 4(e)) in polar air masses far north of the polar front (not shown). The low-level forcing was weaker than during IOP1 and dominated by a polar low-like short-wave trough, which developed in a cold air outbreak south of Svalbard reaching the Norwegian coast at about 6 UTC on 13 December. Mountain wave generation was restricted to northern Scandinavia (Fig. 4(f)) and started at about 0 UTC on 13 December and EF stopped immediately when the polar low dissolved at about 17 UTC on 13 December (Fig. 5(e) and (f)). In the troposphere interfacial waves (Sachsperger et al., 2015) formed at a layer with increased stratification between about 2.5 and 5 km altitude. This layer was the residual of a tropopause fold, which passed over northern Scandinavia the day before (not shown) and is visible in the Scorer parameter maximum at about 2.5 km to 3 km in Fig. 5(h) and in a local maximum of the Brunt-Vaisala frequency in Fig. 6(b). Interfacial waves are visible in this layer by means of potential temperature contour lines, which show wave structures with vertical phase lines. As already the upstream profile was favourable for interfacial waves, wave trapping was stronger during IOP5 than during IOP1.

## 4.2   Wave propagation into the stratosphere

Because of different cross mountain flow during IOP1 and IOP5, the vertical wave propagation in the stratosphere was different. While during IOP5 continuously propagating GWs developed, GW breaking occurred at altitudes between about 25 km to 30 km during IOP1, which is visible from convective overturning, reduced Richardson numbers $Ri$, increased nonlinearity ratios (NLR, Kruse and Smith, 2015) and decreasing energy fluxes in this altitude range during phase P3 in Fig. 5. As breaking mountain waves slow down the background flow (Nappo, 2002) this turbulent region could also be observed by radiosondes started at Andenes, which measured strongly reduced horizontal wind speeds (smaller than 10 m s$^{-1}$) between altitudes of 25 km to 30 km (not shown).

To investigate the reason for different propagation conditions in the stratosphere the NOTOPO and OCEAN simulations are analysed. Wind speeds in these simulations can be regarded as atmospheric background state without perturbations due to mountain waves. The solid lines in Fig 7 show time series of horizontal wind speeds averaged between 25 km to 30 km at Abisko obtained from the NOTOPO and OCEAN simulations and reveal about 10 m s$^{-1}$ lower wind speeds during IOP1 compared to IOP5. The grey shading in Fig. 7 indicates maximum and minimum GW-induced wind speed perturbations obtained by subtracting OCEAN from CTRL run fields. During IOP1 weaker background winds and stronger wind perturbations generated regions with winds below 10 m s$^{-1}$, which favour local mountain wave breaking due to the formation of critical

levels. This means that the growing wave amplitude generates regions with nearly zero winds while the vertical wavelength

approaches zero. This leads to convective overturning and turbulent wave breaking (Dörnbrack, 1998), which is visible in the

regions of reduced Richardson number and increased NLR in Fig. 5(b). During IOP5 wind speeds stayed above 20 m s$^{-1}$ at

this altitude range and enabled wave propagation to altitudes above 30 km (Fig. 5(f)). In addition, the comparison of NOTOPO

and OCEAN simulations shows nearly no difference in horizontal wind speeds. This indicates that GWs are not generated

by changes in roughness lengths in the NOTOPO run, when the flow passes over the ocean and reaches the land surface, as

NOTOPO (water and land surfaces) and OCEAN (only water surfaces) simulations generate a similar wind field.

### 4.3 GWs in the UTLS region

Beside the GW evolution in the lower and middle stratosphere the focus of this study is on GW structures in the UTLS, as

this part of the atmosphere was observed by airborne measurements. Differences in the atmospheric background conditions

during IOP1 and IOP5 are shown by means of average vertical profiles of wind speed, Scorer parameter and vertical energy

flux in Fig. 8. CTRL upstream profiles of horizontal wind speed were averaged over a region between 69°N to 70°N and 10°E

to 15°E (see small black boxes over the ocean in Fig. 4) and indicate relatively strong and constant horizontal wind speeds

between 25 m s$^{-1}$ to 30 m s$^{-1}$ in the troposphere during IOP1. In contrast, a strong increase in wind speed from 10 m s$^{-1}$ near

the surface to 50 m s$^{-1}$ at the tropopause existed during IOP5. Upstream profiles of the Scorer parameter show continuously

increasing values in the troposphere during IOP1 (with a minor local maximum below 2 km altitude), which is not favourable

for the formation of trapped waves. During IOP5 maximum values of $l$ occured at about 2 km altitude and the Scorer parameter

was strongly decreasing between 2 km and 4 km and between 5 km and 8 km altitude during IOP5. This means that during

IOP5 background atmospheric conditions were favourable for the formation of interfacial waves. Regionally averaged profiles

over the mountain region between 67°N to 69°N and 15°E to 25°E (see large black box in Fig. 4) of vertical energy fluxes

show upward propagating waves with relatively constant energy fluxes with height up to 15 km altitude during IOP1 and a

strong peak in energy fluxes in the jetstream region during IOP5.

To simplify the meteorological conditions and to investigate principal differences in wave patterns during IOP1 and IOP5

in the UTLS region, idealised 2D simulations (IOP1ID and IOP5ID) were performed in addition to real-case simulations.

The formation of waves in the idealized 2D simulations (IOP1ID and IOP5ID) is only determined by the upstream wind

profiles, thermal stratification and the mountain peaks at the surface. Effects of horizontal wind shear, convection or 3D wave

propagation are not included. The idealised wave patterns during IOP1 and IOP5 are visualized in Fig. 9. As already seen in

Fig. 8 the situation during IOP1 was characterized by relatively constant horizontal wind speeds with height in the upstream region. Under these conditions hydrostatic waves formed over the Lofoten Islands and the main mountain range with horizontal and vertical wavelengths of 20 km and 6 km and maximum wave amplitudes of 3.3 m s$^{-1}$ and propagated through the low tropopause into the stratosphere. In the lee of the mountains no tropopause fold with interfacial waves as in Fig. 6(a) is visible, because the idealised simulations were initialised with upstream CTRL D3 profiles and the tropopause fold was associated with the synoptic upper-level frontal system low approaching from the north.

During IOP5 the strong jet stream and the stratospheric intrusion layer around 5 km altitude dominated the ambient conditions. Over the mountain range waves with maximum amplitudes of 7 m s$^{-1}$ and horizontal wavelengths of 13 km with nearly vertical phase lines formed between 5 km and 10 km altitude due to the strong increase of horizontal wind speed, which caused the increase of the vertical wavelength. Below 5 km altitude waves with shorter horizontal wavelengths of 6 km is visible. On the eastern side of the mountains interfacial waves with horizontal wavelengths of 8 km formed along the stratospheric intrusion layer and propagated horizontally far in the lee of the mountains. Horizontal wavelengths of interfacial waves in real-case simulations were 10 km (Fig. 6(b)) and therefore in the same order as in the IOP5ID simulation. The extremely stable boundary layer in the lee of the mountains represents a typical situation for the development of resonant trapped lee waves (Sachsperger et al., 2015). The stratification and the Scorer parameter are continuously decreasing with height, which induces the formation of trapped waves with horizontal wavelengths of 6 km in the IOP5ID case (Fig. 9(d)) and 8 km in the CTRL D3 run (Fig. 6(b)). Profiles of idealised EF and MF computed along the cross sections (Fig. 9 (e)) show nearly constant fluxes with height during IOP1. In the IOP5ID case fluxes were very small below an altitude of 5 km due to wave trapping in this altitude range and strongly increased in the jet stream region. Idealised flux profiles are qualitatively similar to fluxes of the CTRL simulations (Fig. 8).

## 5 Observed versus simulated GWs in the UTLS region

### 5.1 Sensitivity of simulated GWs to grid and topography resolution

To study the agreement between observed and simulated wave structures, vertical wind speeds along flight leg 2 of the respective first research flights during IOP1 and IOP5 are shown in Fig. 10 and 11. In all panels the complete flight leg is shown except in (b) where a smaller part of the leg is shown to enlarge the wave structures. Airborne lidar and in-situ measurements in (a) and (b) show alternating up- and downward motions along the flight legs with amplitudes of 2 m s$^{-1}$ to 4 m s$^{-1}$. The

strongest signals occur directly over the mountains and extend horizontally up to 300 km eastward in the lee of the mountain range during both IOP1 and IOP5. White areas in lidar observations mark regions where no measurements are available due to cloud coverage (Witschas et al., 2017). It can be recognized in Fig. 10 and 11 (a) and (b) that waves in the lower troposphere were nearly vertically oriented with weak upstream tilts of the phase lines. During IOP1 (Fig. 10 (b)) the combination of lidar observations with in-situ observations, which were close to the tropopause, enables to identify a stronger phase tilt in the UTLS compared to waves in the lower troposphere. During IOP5 (Fig. 11 (b)) both lidar and in-situ observations were conducted below the tropopause on the shown flight leg, which means that nearly vertical phase lines observed by the lidar continued in the in-situ measurements.

The simulated vertical wind fields in Fig. 10 and 11 depend strongly on the grid and topography resolution. As the CTRL D1 run does not resolve single mountain peaks (cf. Fig. 3), the vertical wind field is large-scale. With a horizontal resolution of 2.4 km and 0.8 km the mountain peaks and related waves are resolved in the CTRL D2 and D3 simulations, however, with weaker maximum amplitudes of 2.4 m s$^{-1}$ and 5.3 m s$^{-1}$ (2.5 m s$^{-1}$ and 4.1 m s$^{-1}$) compared to observed amplitudes of 5.6 m s$^{-1}$ (4.7 m s$^{-1}$) during IOP1 (IOP5) mainly due to numeric diffusion. As in the observations waves show nearly vertical phase lines in the troposphere over the mountains, while upward propagating waves with stronger phase tilts are visible in the stratosphere in the CTRL D2 and D3 runs. The combination of high model resolution with smoothed topography in the SMTOPO D3 simulation (Fig. 10 and 11 (f)) results in a vertical wind field, which is very similar to the coarse resolution CTRL D1 run (Fig. 10 and 11 (c)). This shows that realistic vertical wind fields can only be simulated with a model topography that includes single mountain peaks (Fig. 3). Similar results were found in Leutbecher and Volkert (2000) for GWs over southern Greenland and in Udina et al. (2017), who analysed WRF simulations of a trapped lee wave event over the Pyrénées. In both cases GWs were not captured appropriately in simulations with grid distances larger than 1.3 km and 1 km, respectively due to smoothed topography and numerical diffusion.

On the upstream side of the mountains all simulations reveal no vertical winds at flight level in contrast to lidar and in-situ observations, which show vertical wind perturbations of up to 1.6 m s$^{-1}$ and 0.6 m s$^{-1}$ during IOP1 and IOP5, respectively. This can be explained by missing perturbations in simulations, e.g., due to convective GWs excited further upstream. It is assumed that the east-west extent of the WRF modelling domains (see Fig. 1) would have to be much larger to allow the development of convection induced GWs in the westerly flow over the ocean. This was, however, not possible in this study due to limitations in computational resources. In addition, missing wave structures in the ECMWF analysis data, which are used as initial and boundary conditions for the WRF model, contribute to the smooth upstream vertical wind fields.

The relation between topography resolution and simulated vertical wind field is demonstrated by comparing Fig. 3 and

Fig. 12. The latter shows vertical wind speeds at flight level in (a) and (b) along the two example flight legs during IOP1 and

IOP5 (the same legs as in Fig. 10 and 11). As seen in Fig. 10 and 11 the wave structures cannot be computed in the ECMWF,

CTRL D1 and SMTOPO runs, but occur in the CTRL D2 and D3 simulations with weaker amplitudes over and in the lee of the

mountains compared to observations. The power spectra of observed vertical velocity reveal dominant wavelengths between

15 km to 30 km. Similar wavelengths were obtained from the corresponding topographies in Fig. 3, which indicates that

waves observed in vertical wind fields over the Scandinavian mountains were connected to single mountain peaks. The CTRL

D2 and D3 runs partially reproduce these wavelengths, however, with significantly smaller amplitudes, while they cannot be

resolved by CTRL D1, SMTOPO and ECMWF. Figure 13 shows the horizontal wind component from a direction of 300°,

which represents the flow across the mountains (Dörnbrack and Leutbecher, 2001) along the same flight legs as in Fig. 3 and

Fig. 12. During both IOPs a strong jump in horizontal wind speed is visible in the lee of the mountains (Fig. 13(a), (b)). All

WRF simulations compute very similar horizontal winds independently of the horizontal resolution with maximum deviations

from the in-situ observations of 6.4 m s$^{-1}$ and 8.1 m s$^{-1}$ during IOP1 and IOP5, respectively. In-situ observations indicate

strong fluctuations in the horizontal wind with amplitudes of up to 5 m s$^{-1}$ and 3.6 m s$^{-1}$ during IOP1 and IOP5. In contrast

to the vertical wind, the spectra of the horizontal wind component are dominated by larger wavelengths of 60 km to 120 km.

In addition, power spectra of in-situ observations show secondary maxima for smaller wavelengths of about 25 km to 40 km,

which are also computed by the CTRL D2 and CTRL D3 simulations during IOP5. The good agreement of horizontal winds

and related spectra between all CTRL and SMTOPO simulations indicates that horizontal winds are less dependent on the grid

and the topography resolution compared to the waves observed in vertical wind fields.

## 5.2    Observed versus simulated energy and momentum fluxes

Airborne observations during IOP1 and IOP5 enable to verify the simulation of GW-induced energy and momentum fluxes in

the UTLS for different grid resolutions. EF and MF at flight level are computed according to the method of Smith et al. (2008,

see section 3.3) and are shown in Fig. 14. During both IOP1 and IOP5 the linear Eliassen-Palm relation between EF and MF

is satisfied nearly perfectly in all WRF simulations and indicates upward propagating mountain waves. Observed EF and MF

values show, however, an offset from the identity line and include negative values during IOP5 Fig. 14(b), which means that

non-linear effects seem to be underestimated by the WRF model. The Eliassen-Palm relation was fulfilled already in other

GW campaigns, e.g., over the Sierra Nevada (Smith et al., 2008) and New Zealand (Smith et al., 2016). Note that only data

directly above the mountains (cf. black dotted lines in Fig. 2) are used for flux computations to avoid nonlinearity effects in observational data in upstream regions. For ECMWF the relatively short lengths of the mountain legs cause inadequate linear Eliassen-Palm relations. Along complete legs (cf. black dashed lines in Fig. 2) the linear relation is achieved well for ECMWF (not shown).

Surprisingly, energy and momentum fluxes are significantly larger in the WRF CTRL runs during both IOPs compared to in-situ observations (up to 10 W m$^{-2}$ during IOP5) in spite of smaller wave amplitudes (cf., Fig. 10 and 11). As already seen in Fig. 8 fluxes were strongest for D3 simulations due to higher vertical wind speeds compared to D1 and D2 simulations (cf. Fig. 10 and 11).

To study the reason for increased fluxes in the model, sensitivity runs with an increased vertical grid resolution of 80 m (CTRLVR) and increased horizontal and vertical turbulent diffusion (HVDIFF, H2VDIFF) were performed (see Table 2). The idea of these sensitivity experiments was to improve the representation of small-scale non-linear effects like wave breaking, which reduces vertical wind speeds and contributes to a reduction of the energy and momentum fluxes. By comparing EF and MF from different simulations, Fig 14 shows that an increased vertical grid resolution (CTRLVR) slightly reduces the EF and MF values in the order of 2 W m$^{-2}$. Probably, an additional increase of the horizontal resolution towards the order of a large-eddy simulation (LES) would be necessary to reduce the simulated fluxes by explicitly resolving wave breaking. By switching on horizontal turbulent diffusion in the HVDIFF simulation vertical fluxes were reduced by about 2 W m$^{-2}$, which is similar to the CTRLVR simulation. A clear improvement of vertical flux computation was attained by both applying horizontal diffusion and doubling the vertical mixing tendency term obtained from the boundary layer parameterization scheme in the H2VDIFF simulation. The propagation of wave energy was effectively damped by up to 6 W m$^{-2}$ compared to the CTRL run.

Grey dots in Fig. 14 mark smoothed observed fluxes (In-situSM), which were computed by using horizontal wavelengths larger than 15 km. These fluxes are nearly identical to the original in-situ fluxes, which means that waves with wavelengths smaller than 15 km did not contribute significantly to GW fluxes at flight level. Similar results were found by Smith et al. (2016) for "fluxless" waves with wavelengths between 6 km to 15 km over New Zealand during the DEEPWAVE campaign.

Profiles of EF and MF of D3 simulations, which were averaged over all flight legs during IOP1 and IOP5, respectively are shown in Fig. 15. For the CTRLVR, HVDIFF and H2VDIFF simulations EF and MF fluxes are reduced over the atmospheric column between 2 km and 15 km and agree better with in-situ observations than fluxes obtained from CTRL runs. CTRLVR and HVDIFF simulations indicate similar flux profiles, while the H2VDIFF runs show clearly reduced fluxes. During IOP5 the largest improvement of the H2VDIFF fluxes compared to CTRL run fluxes can be found in the lower troposphere at altitudes

between about 2.5 km and 7.5 km. This is at altitudes of the layer where wave trapping occured (see Fig. 6) and localized regions of wave breaking increased turbulent mixing. These processes seemed to be underestimated in the CTRL simulations.

## 5.3 Model verification

In order to verify the model results of the previous sections, CTRL simulations are compared quantitatively to in-situ and lidar observations. Figure 16 shows correlations of airborne in-situ and lidar measurements with numerical models for potential temperature, wind direction and vertical and horizontal wind speed for all legs during IOP1 and IOP5. Except vertical wind speed all variables are captured well by both the WRF and ECMWF model with similar correlation coefficients of up to 0.99 and root mean square errors (RMSE) independently of the horizontal resolution. This good agreement can be explained as fields of potential temperature and horizontal wind speed are principally dominated by large-scale waves (cf. Fig. 13). Vertical winds on the other hand reflect small-scale up- and downdrafts (Witschas et al., 2017), which are linked to single mountain peaks (cf. Fig. 3 and Fig. 12) and may be shifted slightly in space and time in the models, which complicates a pointwise comparison with measurements. As ECMWF is a hydrostatic model, vertical velocity is a diagnostic variable and GW-induced vertical winds cannot be resolved, which results in very low correlation coefficients in Fig. 16(d) and (f). A separation of correlation coefficients for both IOPs is listed in Table 3 and indicates that IOP5 was captured better by the models than IOP1 probably due to the less complex meteorological situation (see section 4).

To verify vertical velocities in a different way, the distribution of lidar, CTRL and ECMWF vertical winds are computed. Figure 17 (a) shows the distribution of vertical velocity along all flight legs during both IOP1 and IOP5, where the lidar was operating in nadir pointing mode. The observed lidar data exhibit a broad distribution with large wave amplitudes of maximum up- and downdrafts of 5.0 m s$^{-1}$ and -8.1 m s$^{-1}$. The CTRL D3, D2, D1 and especially the ECMWF model simulate narrower distributions with maximum and minimum vertical winds of 8.25 m s$^{-1}$ and -8.23 m s$^{-1}$, 4.7 m s$^{-1}$ and -5.3 m s$^{-1}$, 0.99 m s$^{-1}$ and -0.95 m s$^{-1}$ and 0.46 m s$^{-1}$ and -0.35 m s$^{-1}$, respectively.

Figure 17 (b) shows the relation between the mean vertical velocity amplitude along all nadir pointing lidar flight legs and the horizontal model resolution (for lidar data a resolution of 800 m was applied; see section 2.2). The largest improvement in simulating vertical velocities is achieved by reducing the horizontal mesh size from 7.2 km to 2.4 km (CTRL D1 and CTRL D2) due to the more realistic representation of the topography in CTRL D2 (see Fig. 3). The importance of a properly resolved topography for the simulated vertical wind field is further indicated by the SMTOPO runs, which show nearly the same amplitudes as the CTRL D1 run independently of the model grid resolution. Equal values of the NOTOPO and OCEAN

simulations indicate that GWs are not induced by a change in roughness length when the flow passes the coast line in the NOTOPO simulation. The more realistic computation of EF and MF in the HVDIFF and H2VDIFF simulations due to increased turbulent diffusion results in reduced vertical wind speeds of up to $0.1$ m s$^{-1}$ on average, while higher vertical grid resolutions in the CTRLVR simulations did not change vertical wind fields significantly.

## 6   Conclusions

In this study two mountain wave events were analysed, which occured during the GW-LCYCLE I field campaign in December 2013 by means of airborne observations and numerical simulations. During the campaign the DLR Falcon was stationed at Kiruna airport to measure GWs above northern Scandinavia. Airborne in-situ and lidar observations were accompanied by ground-based lidar, radar and radiosonde observations on the wind- and leeward side of the Scandinavian mountain range. In contrast to Ehard et al. (2016), who analysed the same GW cases with a focus on waves in the middle atmosphere, this paper concentrated on GW structures in the tropo- and lower stratosphere.

During both events the situation was dominated by westerly cross-mountain flow with different atmospheric upstream conditions, which induced variable GW development over and in the lee of the mountains. Weaker stratospheric winds during IOP1 caused GW breaking between 25 km to 30 km altitude compared to deeper GW propagation during IOP5 (cf., Ehard et al., 2016). In the troposphere, a stratified layer at 5 km altitude formed favourable conditions for the generation of interfacial waves during IOP5. During IOP1 upstream conditions were not conducive for wave trapping, but a synoptic tropopause fold on the eastern side of the mountains enabled weak wave trapping in the CTRL simulations.

A large number of numerical simulations were performed to test the ability of a state of the art mesoscale model to capture the meteorological situation and to properly simulate the observed small-scale GWs. A special focus was on the correct representation of vertical winds and GW-induced vertical energy and momentum fluxes. Observations and simulations showed that up- and downdrafts had a strong linkage to single mountain peaks and horizontal wavelengths obtained from vertical winds were in the order of 15 km to 30 km. Wave structures deduced from horizontal wind speeds were dominated by larger wavelengths between 60 km to 120 km and represented GWs excited by the main mountain range. The intercomparison of numerical simulations revealed that wave structures in horizontal winds were captured well by all model runs nearly independently of the horizontal grid resolution. The analysis of vertical wind fields exhibited that single mountain peaks must be represented

correctly in the model topography and that a horizontal model grid resolution of at least 2.4 km is necessary over Scandinavia to compute realistic vertical winds.

The calculation of energy and momentum fluxes along all flight legs of the four research flights during IOP1 and IOP5 indicated that the linear Eliassen-Palm relation (Eliassen and Palm, 1960) was satisfied very well especially in the model runs. The completion of this relation was already found in other GW campaigns (e.g., Smith et al., 2008, 2016) During GW-LCYCLE I simulated fluxes were generally larger than observed values (up to 10 W m$^{-2}$ during IOP5) and this discrepancy was most distinct for simulations with high horizontal model grid resolutions due to better resolved vertical winds (cf., CTRL D1 and

CTRL D3). Sensitivity runs demonstrated that simulated fluxes could be reduced by up to 2 W m$^{-2}$ by increasing the vertical grid resolution from about 160 m to 80 m (CTRLVR) and by switching on horizontal turbulent diffusion (HVDIFF). A reduction of up to 6 W m$^{-2}$ was achieved by activating horizontal diffusion and additionally doubling the tendency terms computed by the boundary layer scheme (H2VDIFF), i.e., intensifying the effect of vertical turbulent mixing in regions of GW breaking. In all three cases small-scale non-linear effects like GW breaking were amplified, which damped the vertical propagation of

waves and related energy and momentum fluxes. This result makes clear that quasi-linear wave propagation dominated in the presented simulations even for small grid distances of 800 m (CTRL D3) and that the used boundary layer scheme underestimated turbulent mixing induced by GW breaking. A systematic test of further boundary layer parameterizations would be necessary to study if other schemes produce similar results. Further investigations could focus on disagreements between simulated and observed GWs on the upstream side of the mountains, which were not included in ECMWF and WRF simulations,

but strongly disturbed in observations. WRF runs driven by ECMWF ensemble members could be a first step to investigate the role of upstream variability on the resulting GW structures.

*Acknowledgements.* This study was funded by the German ministry of research in the framework of the projects "Processes and climatology of gravity waves" (PACOG, grant RA 1400/6-1) and "Modification of gravity waves propagating across the tropopause (GW-TP, grant DO 1020/9 1)" within the research unit "Multiscale dynamics of gravity waves" (MS-GWaves) and by the project "Role of the middle atmosphere

in climate" (ROMIC) under grant 01LG1206A. Mesoscale simulations were performed at the Leibniz Institute for Atmospheric Physics (IAP) in Kühlungsborn, Germany. The code to compute the diagnostic GW energy and momentum fluxes was kindly provided by Chris Kruse, Yale University.

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

580

**Table 1.** Overview of intensive observation periods (IOP) during GW-LCYCLE I. Start and end times of research flights are indicated in UTC. For each flight the number of airborne lidar profiles in both nadir and scanning mode (see section 2.2) and the corresponding coverage of useable data in percentage is shown. Radiosondes were released at Andenes (A), Esrange (E), Kiruna (K) and Sodankylä (S) (see Fig. 1).

| IOP | Date | Research flights | | Airborne lidar profiles | | Radiosondes | | | | Description |
|---|---|---|---|---|---|---|---|---|---|---|
| | | **Start** | **End** | **Nadir** | **Scan** | **A** | **E** | **K** | **S** | |
| **1** | 03.12.2013 | 09:13 | 11:26 | 2379 (41.93%) | 6 (26.24%) | 9 | 9 | 7 | 10 | Mountain wave event |
| | | 13:25 | 16:22 | 1328 (39.24%) | 100 (67.56%) | | | | | |
| 2 | 05.12.2013 | 08:33 | 11:48 | 7222 (36.24%) | 21 (36.85%) | - | - | - | - | Storm Xaver |
| 3 | 09.12.2013 | 12:08 | 14:58 | 2408 (7.84%) | - | - | - | - | - | Trace gas/pollution event |
| 4 | 11.12.2013 | - | - | - | - | 6 | 7 | 7 | 11 | Mountain wave event |
| **5** | 13.12.2013 | 06:10 | 09:37 | 2250 (38.33%) | 68 (53.72%) | 5 | 5 | 8 | 8 | Mountain wave event |
| | | 12:19 | 15:24 | 2305 (47.97%) | 39 (44.19%) | | | | | |

**Table 2.** Overview of real-case simulations for cases with full (CTRL), smoothed (SMTOPO) and without (NOTOPO) topography and for cases with a flat water surface (OCEAN) and increased vertical grid resolution (CTRLVR). Horizontal turbulent diffusion (H) is switched on in the HVDIFF case and vertical diffusion (V) is doubled in the H2VDIFF case.

| Case | Type | Topo. D1 | Topo. D2 | Topo. D3 | Landuse | Diffusion | Vert. resolution (m) |
|---|---|---|---|---|---|---|---|
| CTRL | real-case | full | full | full | land-ocean | V | 80-300 |
| SMTOPO | real-case | full | smoothed | smoothed | land-ocean | V | 80-300 |
| NOTOPO | real-case | flat | flat | flat | land-ocean | V | 80-300 |
| OCEAN | real-case | flat | flat | flat | ocean | V | 80-300 |
| CTRLVR | real-case | full | full | full | land-ocean | V | 80 |
| HVDIFF | real-case | full | full | full | land-ocean | H+V | 80-300 |
| H2VDIFF | real-case | full | full | full | land-ocean | H+2V | 80-300 |

**Table 3.** Correlation coefficients between airborne in-situ and lidar observations and numerical models for potential temperature Θ, horizontal

wind speed $U$, vertical wind speed $w$ and wind direction $dd$ along all flight legs during both IOP1 and IOP5 (numbers in bold).

| Model | In-situ Θ | In-situ $U$ | In-situ $w$ | In-situ $dd$ | Lidar $U$ | Lidar $w$ |
|---|---|---|---|---|---|---|
| CTRL D3 | 0.992 **0.996** | 0.877 **0.963** | 0.154 **0.343** | 0.892 **0.868** | 0.755 **0.917** | 0.242 **0.408** |
| CTRL D2 | 0.993 **0.996** | 0.872 **0.964** | 0.169 **0.361** | 0.883 **0.858** | 0.735 **0.913** | 0.300 **0.417** |
| CTRL D1 | 0.993 **0.997** | 0.872 **0.964** | 0.156 **0.213** | 0.868 **0.858** | 0.712 **0.916** | 0.205 **0.218** |
| ECMWF | 0.992 **0.998** | 0.891 **0.967** | 0.116 **0.076** | 0.858 **0.897** | 0.414 **0.882** | -0.064 **-0.092** |

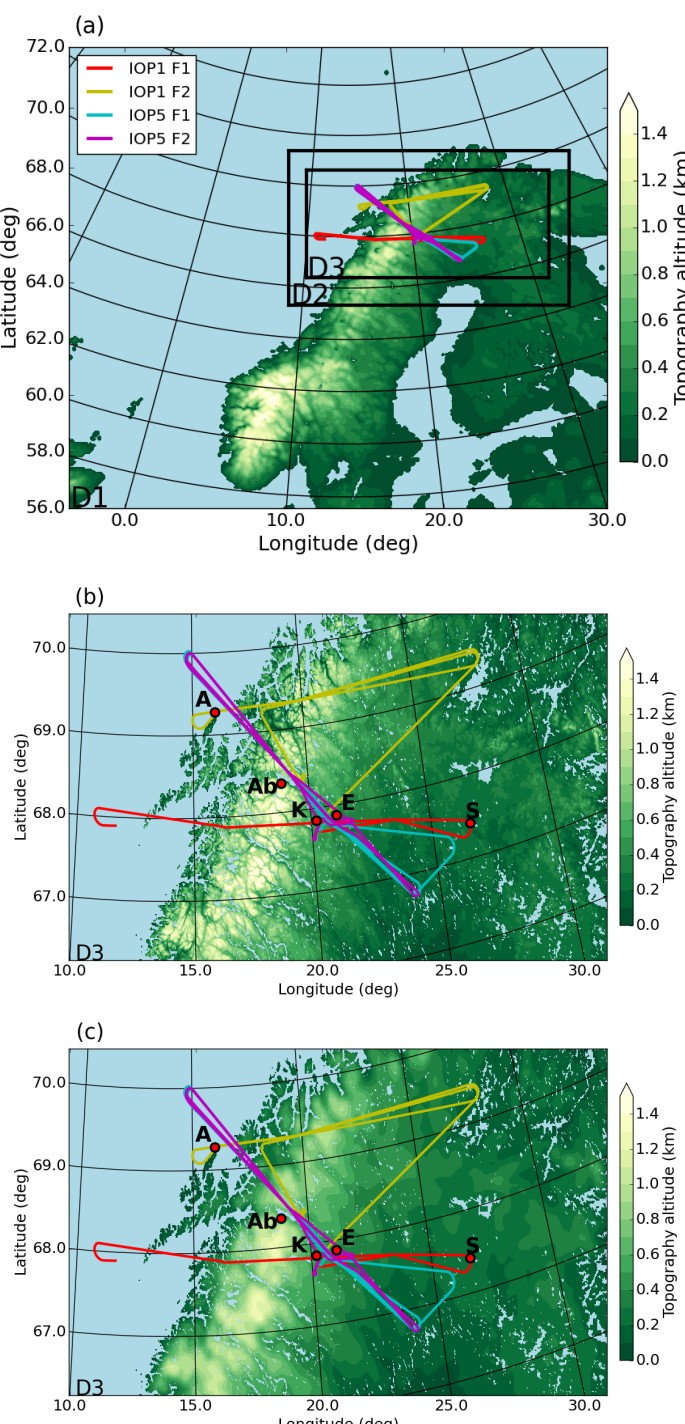

**Figure 1.** Topographic maps of Scandinavia and operational areas of the GW-LCYCLE I campaign. The coloured lines indicate DLR Falcon flighttracks during IOP1 and IOP5. In (a) the shown region and the black boxes mark the modelling domains for mesoscale WRF simulations with horizontal grid sizes $\Delta x$ of 7.2 km (D1), 2.4 km (D2) and 0.8 km (D3), respectively. The topography of domain D3 is shown in (b) for the CTRL and in (c) for the SMTOPO simulations (see Table 2). The red dots mark the position of Andenes (A), Abisko (Ab), Kiruna (K), Esrange (E) and Sodankylä (S).

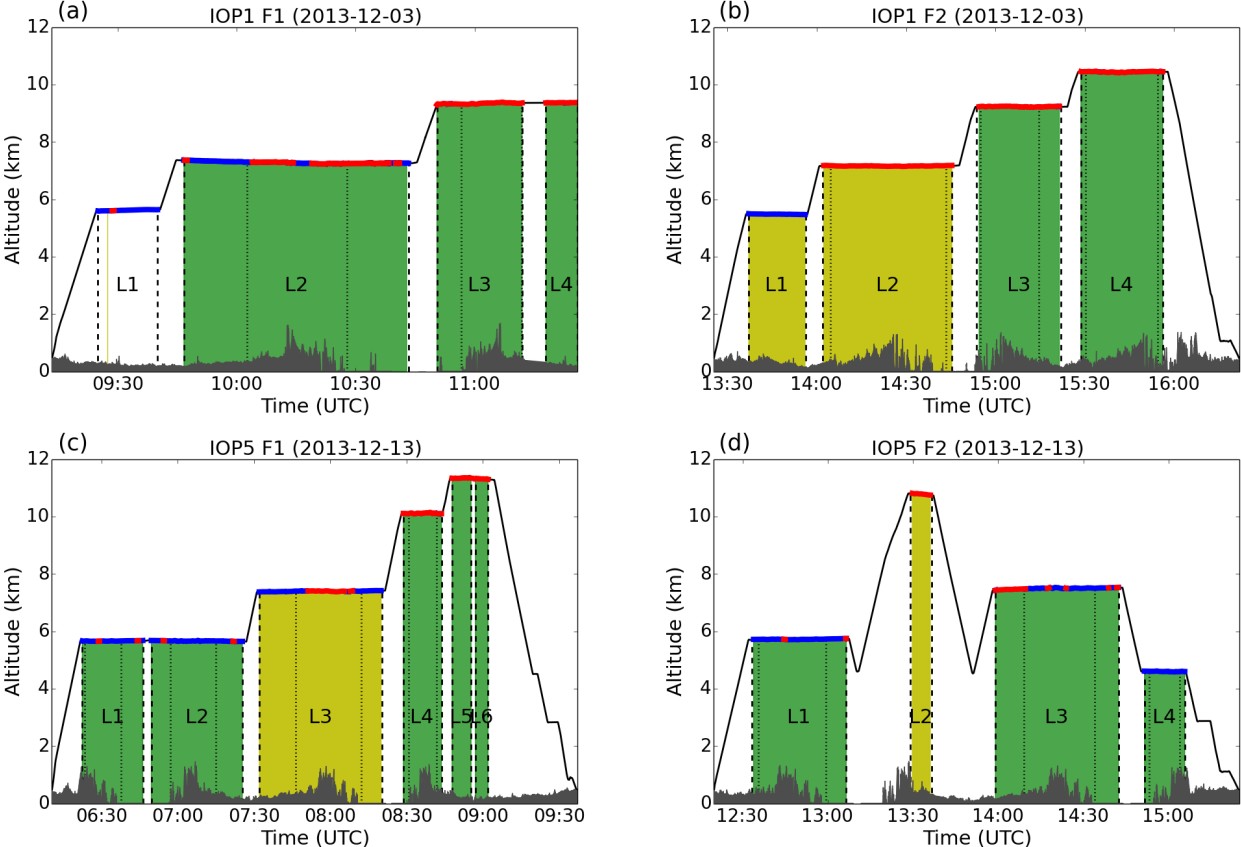

**Figure 2.** Flight legs and altitudes of the four research flights during (a) and (b) IOP1 and (c) and (d) IOP5. The yellow and green shaded areas indicate regions where the airborne lidar operated in scanning and nadir pointing mode, respectively. Blue and red thick lines indicate respective flight altitudes below and above the tropopause, which was determined by in-situ trace gas measurements of $N_2O$ with a threshold value of 326.6 ppbv (see section 2.2). The black dashed and dotted lines mark flight legs used for data analysis and indicate complete flight legs and leg sections limited to mountain regions, respectively. The topography along the flight tracks is based on the ASTER data set.

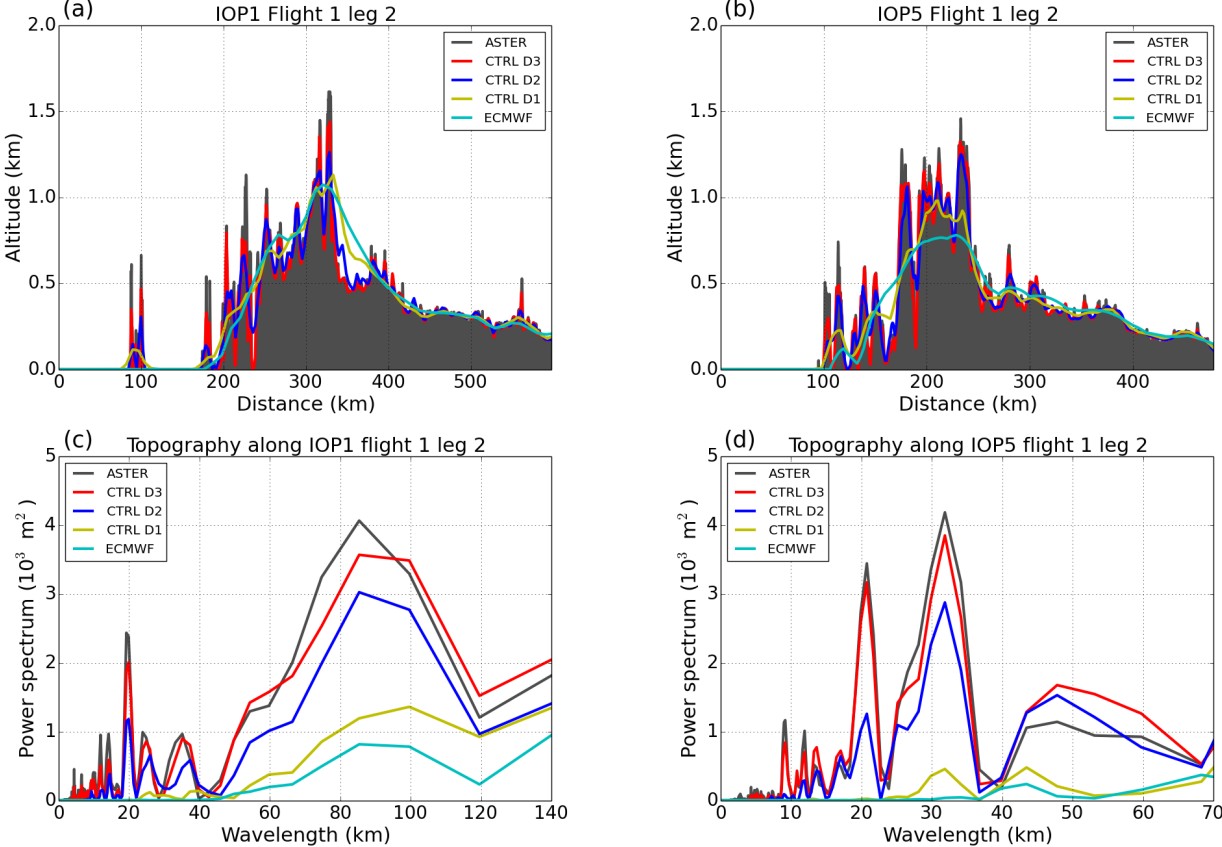

**Figure 3.** Comparison of topography along leg 2 of flight 1 during (a) and (c) IOP1 and (b) and (d) IOP5 for the high resolution ASTER data

set, CTRL and ECMWF topographies. In (c) and (d) powerspectra of the topographies in (a) and (b) are shown.

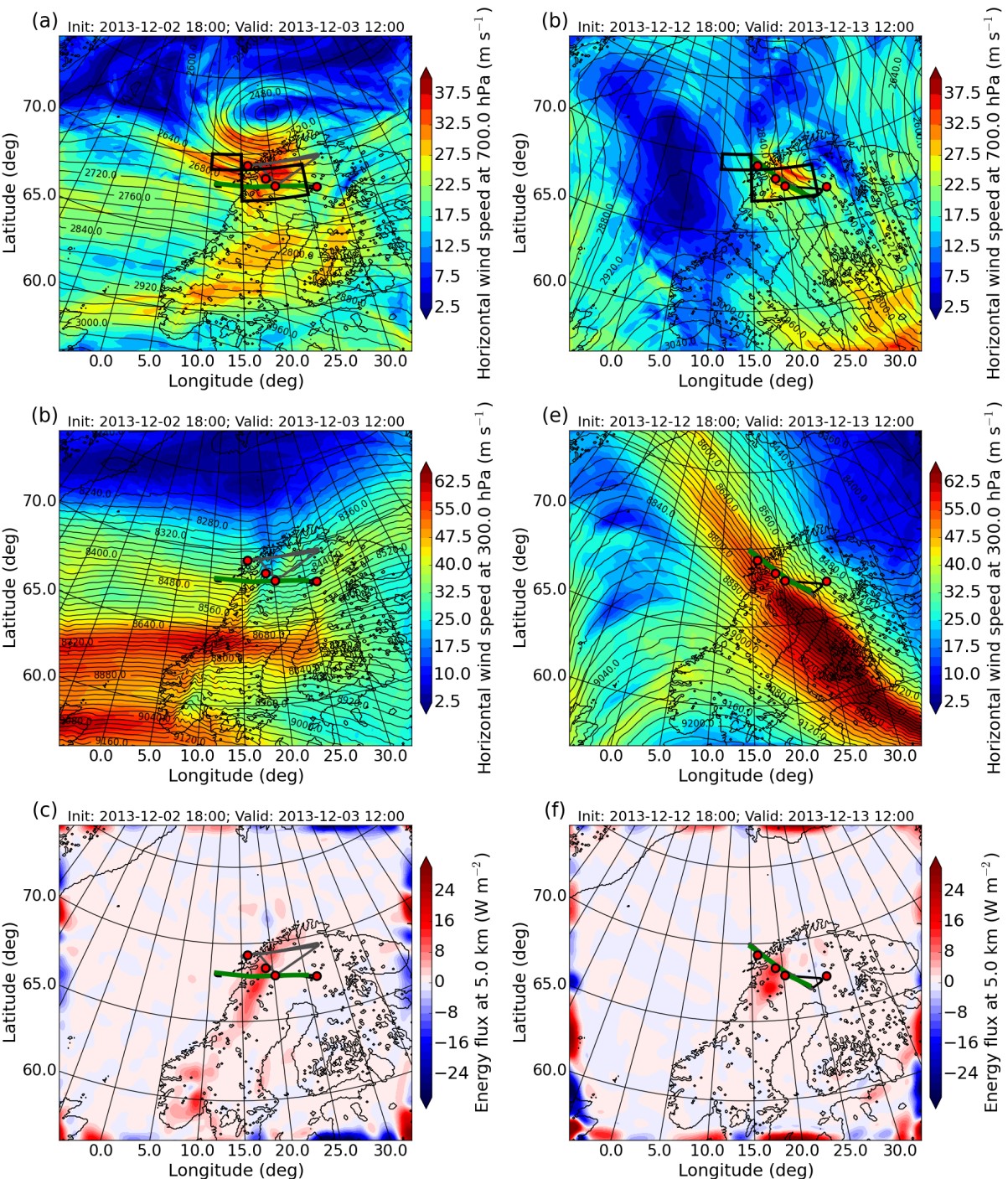

**Figure 4.** Synoptic situation and respective flight tracks during (a) to (c) IOP1 and (d) to (f) IOP5. Shown are CTRL D1 ($\Delta x = 7.2$ km) horizontal wind speed and geopotential height at 700 hPa in (a) and (d) and 300 hPa in (b) and (e). GW-induced vertical energy fluxes are plotted at 5 km altitude in (c) and (f). The red dots mark the locations of Andenes, Abisko, Kiruna and Sodankylä (cf. Fig. 1). The first and second flight of IOP1 and IOP5 are plotted with black and grey lines, respectively. The example flight legs used in this study are marked with a green line. The chosen times at 12 UTC, 3 December and 12 UTC, 13 December are valid between the respective two research flights (cf. Table 1 and Fig. 2). The areas marked with black lines in (a) and (b) indicate regions used for the computation of averaged vertical profiles in Fig. 8.

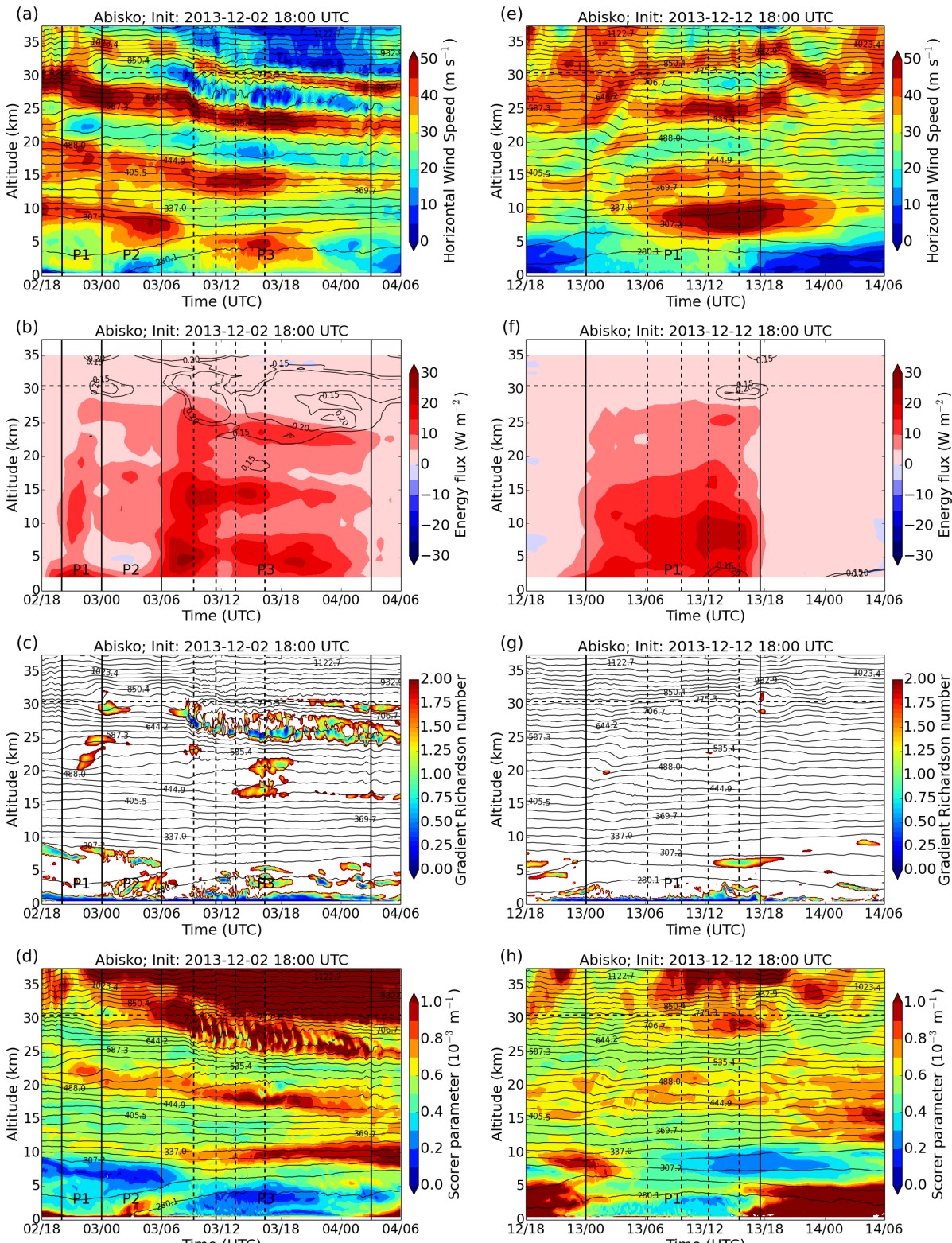

**Figure 5.** Vertical time series of CTRL D2 simulations at Abisko (68° N, 19° E) during (a) to (d) IOP1 and (e) to (h) IOP5 for horizontal wind speed, vertical energy flux, gradient Richardson number and Scorer parameter. The vertical solid and dashed lines mark the different mountain wave phases and periods of research flights. The dashed horizontal line indicates the height of the sponge layer at the model top. Thin black contour lines mark the nonlinearity ratio (NLR) of Kruse and Smith (2015) in (b) and (f) and isentropes in the remaining figures.

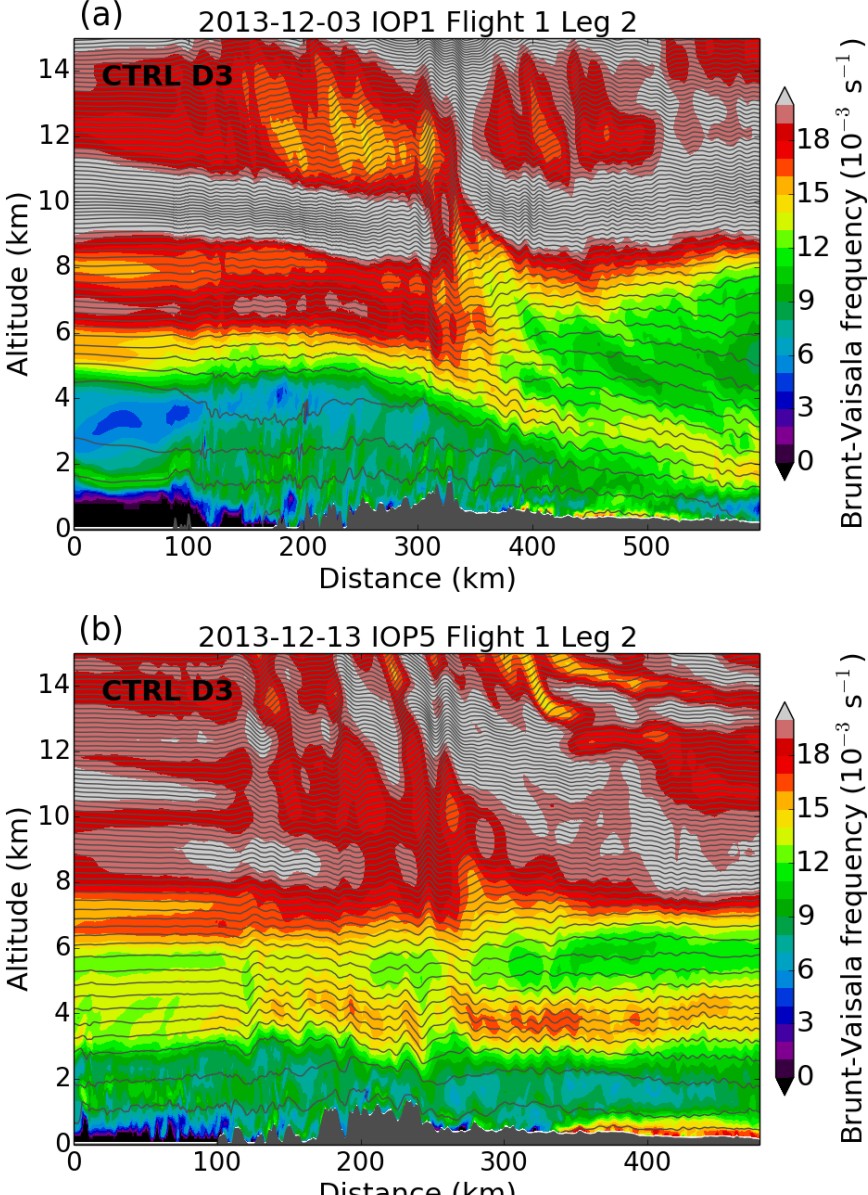

**Figure 6.** Cross sections of CTRL D3 Brunt-Vaisala frequency (coloured contours) of flight 1, leg 2 during (a) IOP1 and (b) IOP5. Potential temperature is indicated with black contour lines.

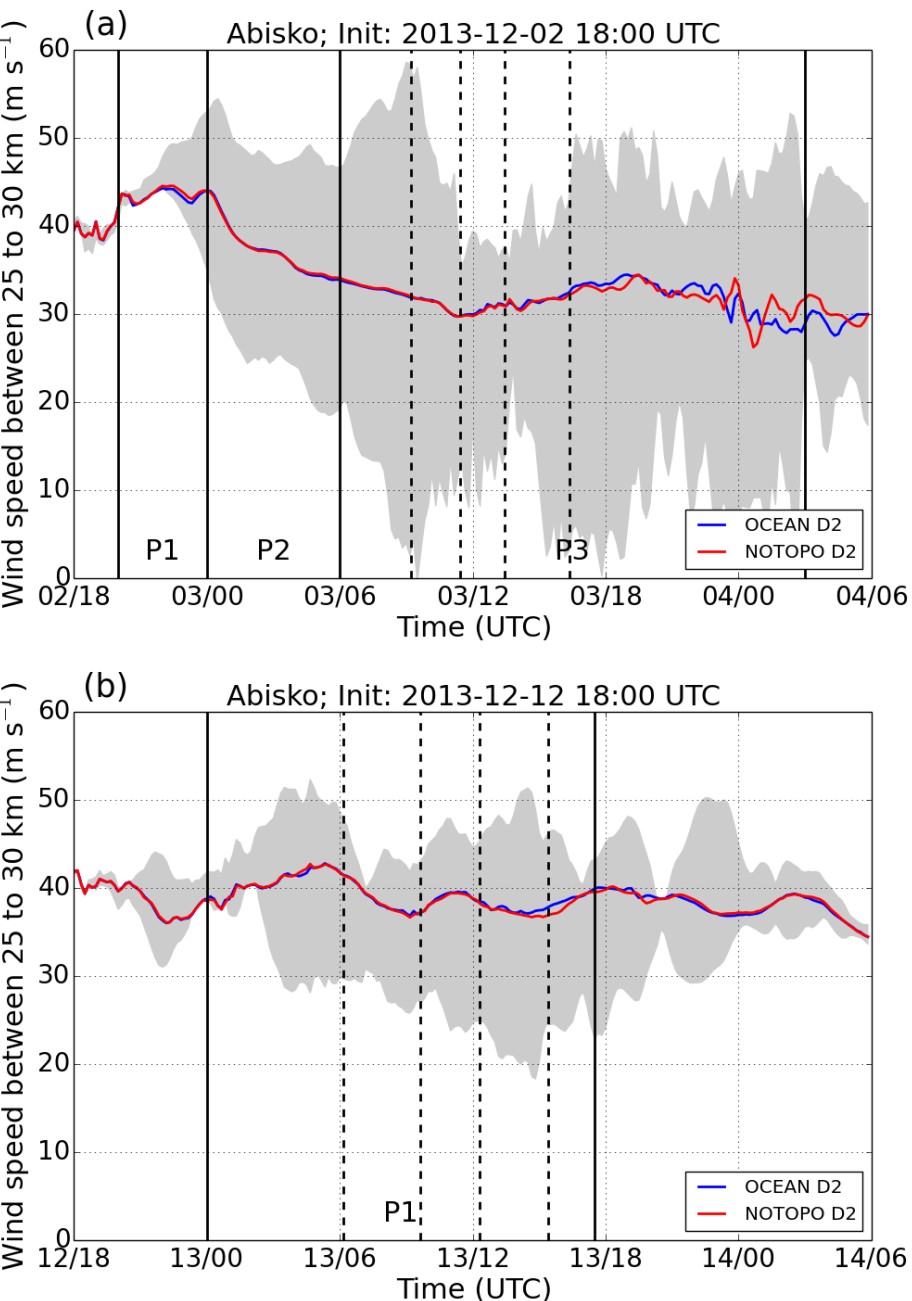

**Figure 7.** Time series of background horizontal wind speed averaged between 25 to 30 km (solid lines) of the OCEAN and NOTOPO simulation at Abisko (68° N, 19° E) for (a) IOP1 and (b) IOP5. The grey shaded area marks the range of minimum and maximum wind speed perturbations (mountain wave induced) at 25 to 30 km determined by subtracting OCEAN from CTRL simulation fields. All data is based on domain D2. The vertical solid and dashed lines mark the different mountain wave phases and periods of research flights as in Fig. 5.

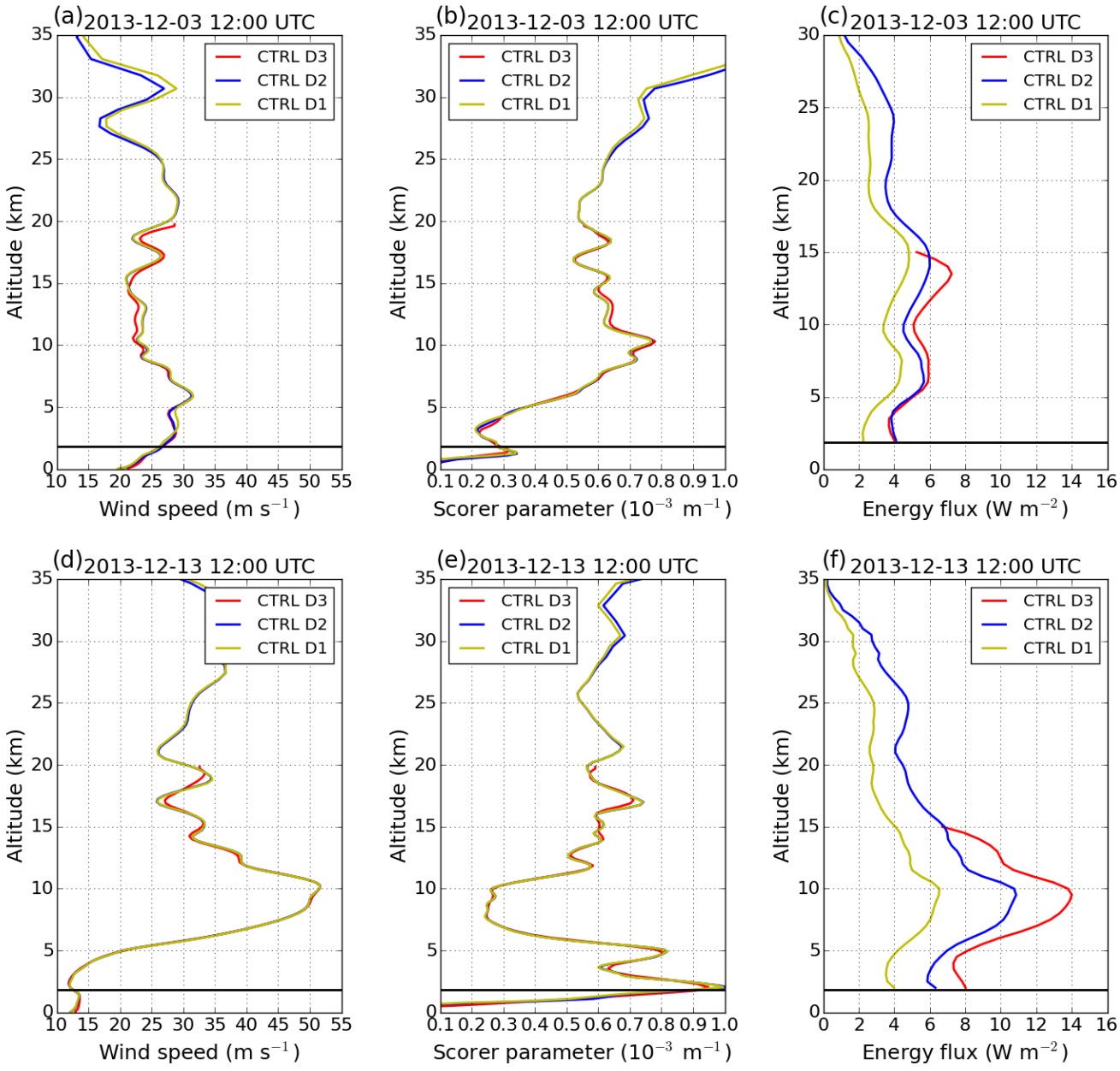

**Figure 8.** Area averaged vertical profiles during (a) to (c) IOP1 and (e) to (f) IOP5. Cross mountain horizontal wind speed (from a direction of 300°) and Scorer parameter are averaged over the upstream area from 69°N to 70°N and 10°E to 15°E, while vertical energy fluxes are averaged over the mountain area within 67°N to 69°N and 15°E to 25°E (see black boxes in Fig. 4(a) and (b)). The thick horizontal black line marks the maximum mountain peak height within the mountain box and the dashed vertical black line in (b) and (e) indicates a horizontal wavelength of 30 km.

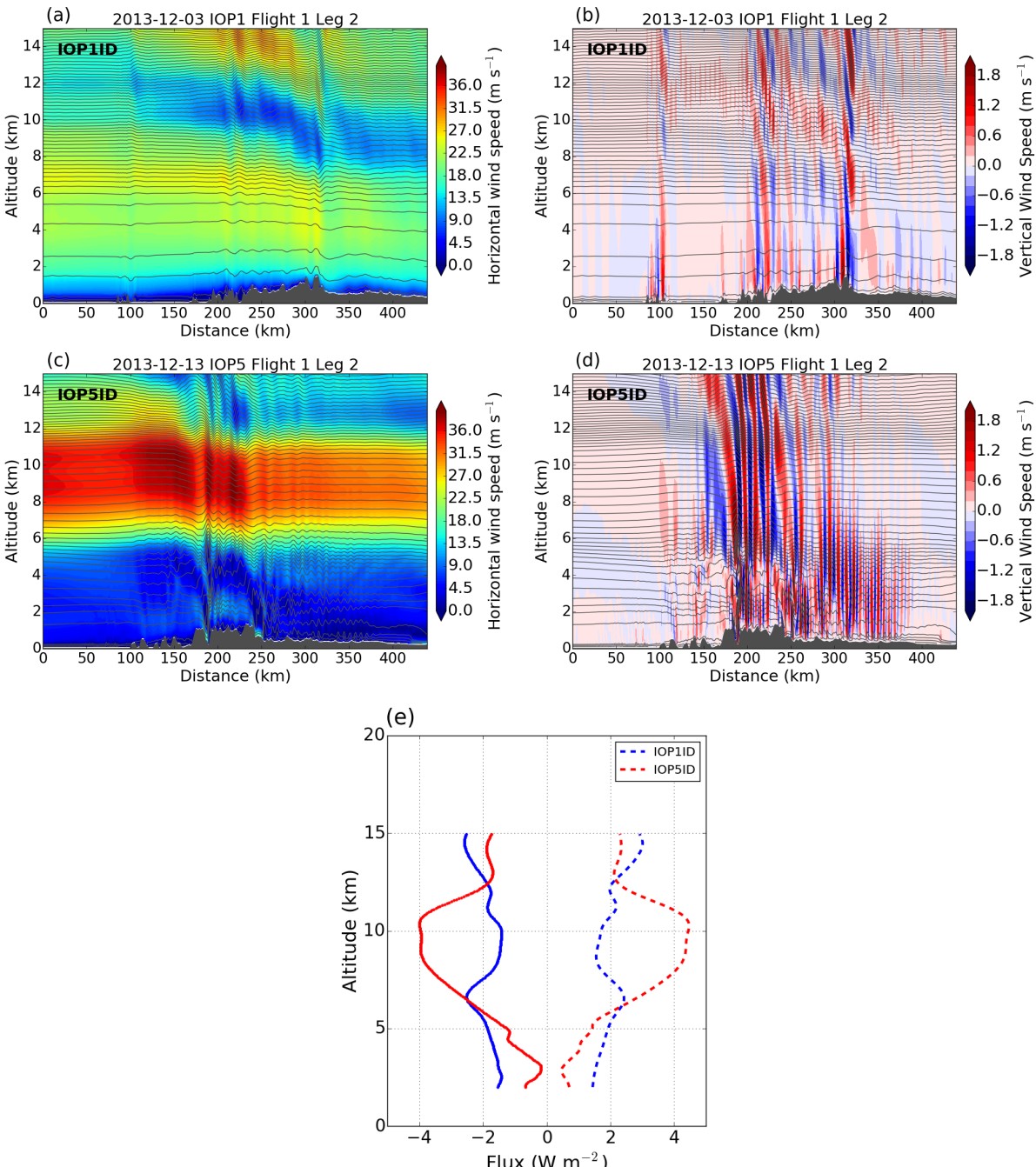

**Figure 9.** Cross sections of idealised 2D simulations (IOP1ID and IOP5ID) for horizontal and vertical wind speed along leg 2 of flight 1 during (a) and (b) IOP1 and (c) and (d) IOP5. Profiles of EF (dashed lines) and MF (dotted lines) are plotted in (e). Simulations were initialized with upstream profiles shown in Fig. 8 for horizontal wind speed.

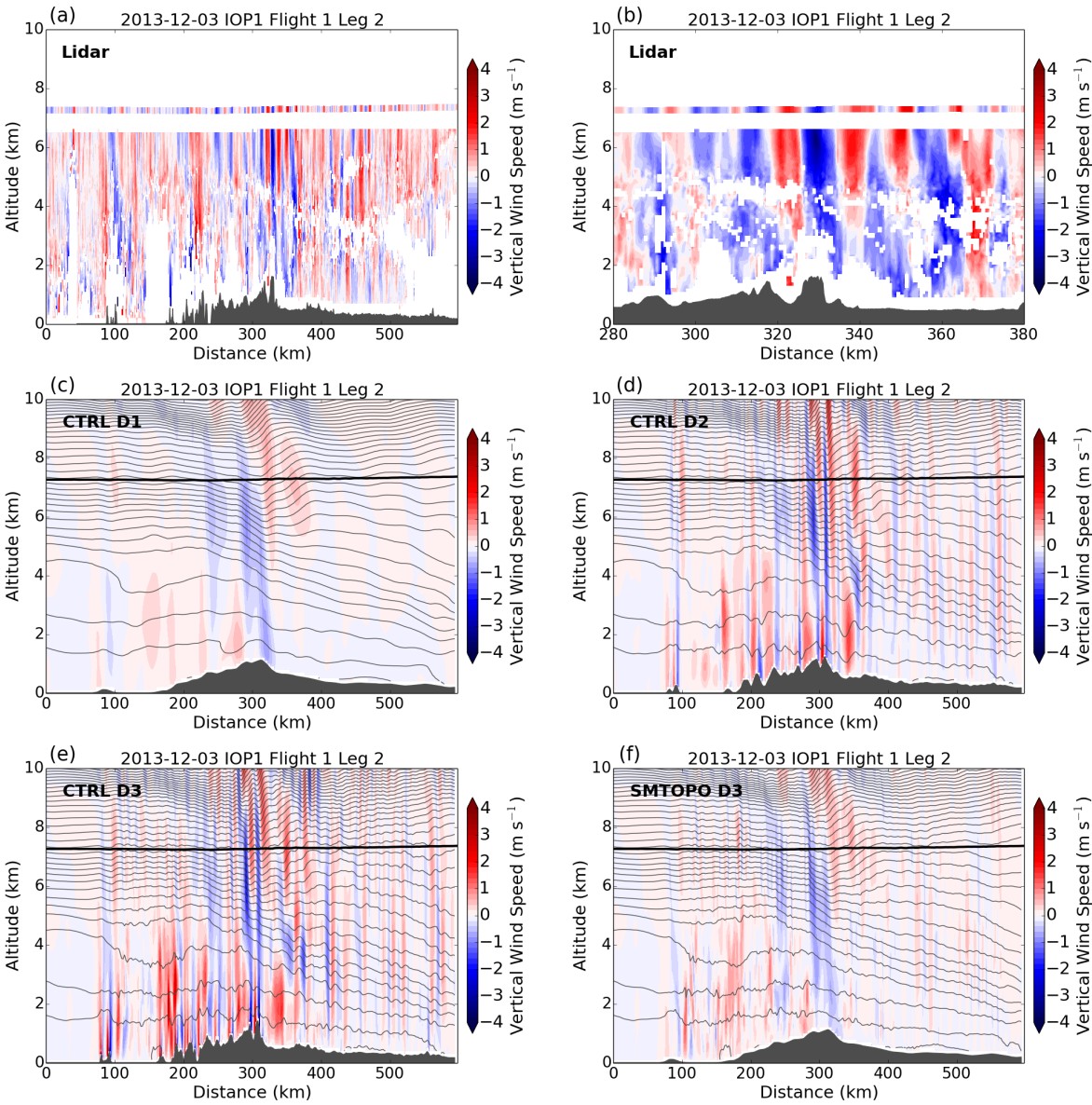

**Figure 10.** Cross sections for vertical velocity of flight 1, leg 2 during IOP1. Lidar and in-situ measurements (data at 7.3 km) are shown in (a) for the complete flight leg and in (b) for a smaller region over the mountains to enlarge the wave structures. Model results for vertical wind and potential temperature (contour interval 2 K) of the CTRL D1, CTRL D2, CTRL D3 and SMTOPO D3 simulations are shown in (c) to (f). The thick black line in (c) to (f) marks the flight altitude.

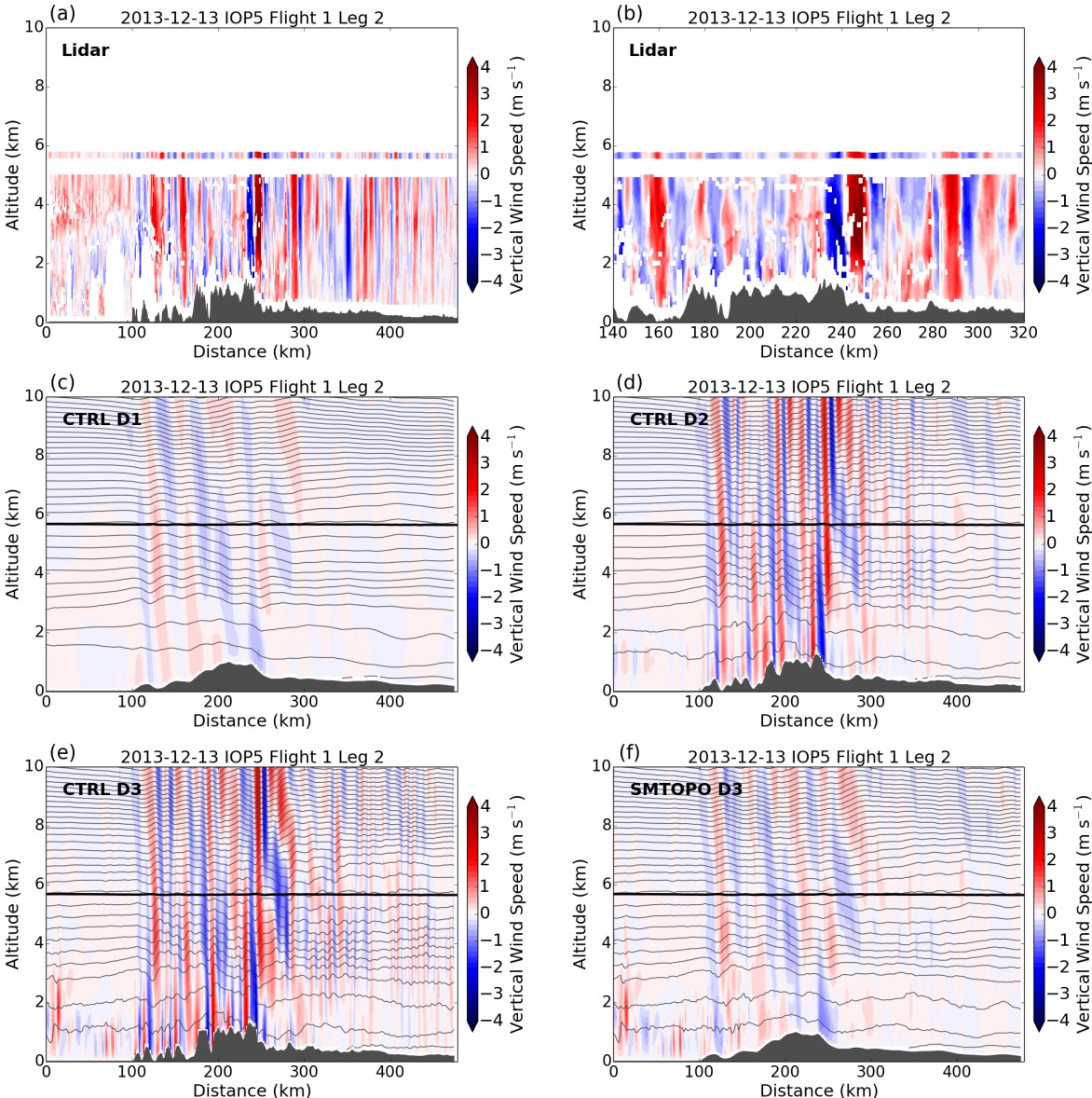

**Figure 11.** As in Fig. 10, but for flight 1, leg 2, during IOP5. The flight level was at 5.6 km.

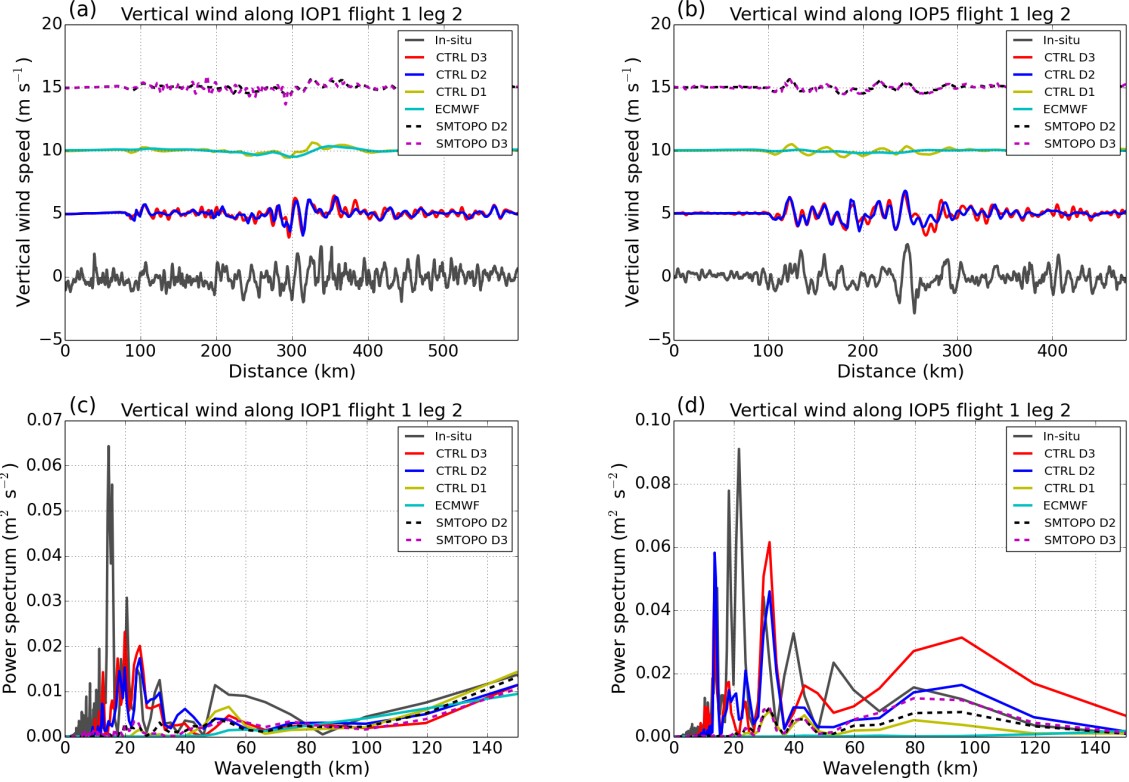

**Figure 12.** Comparison of vertical winds at flight level along leg 2 of flight 1 during (a) and (c) IOP1 and (b) and (d) IOP5. To improve readability, vertical wind speeds are shifted by 5 m s$^{-1}$ in (a) and (b). In (c) and (d) power spectra of vertical winds in (a) and (b) are shown. The corresponding spectra of the topography along the flight legs are plotted in Fig. 3.

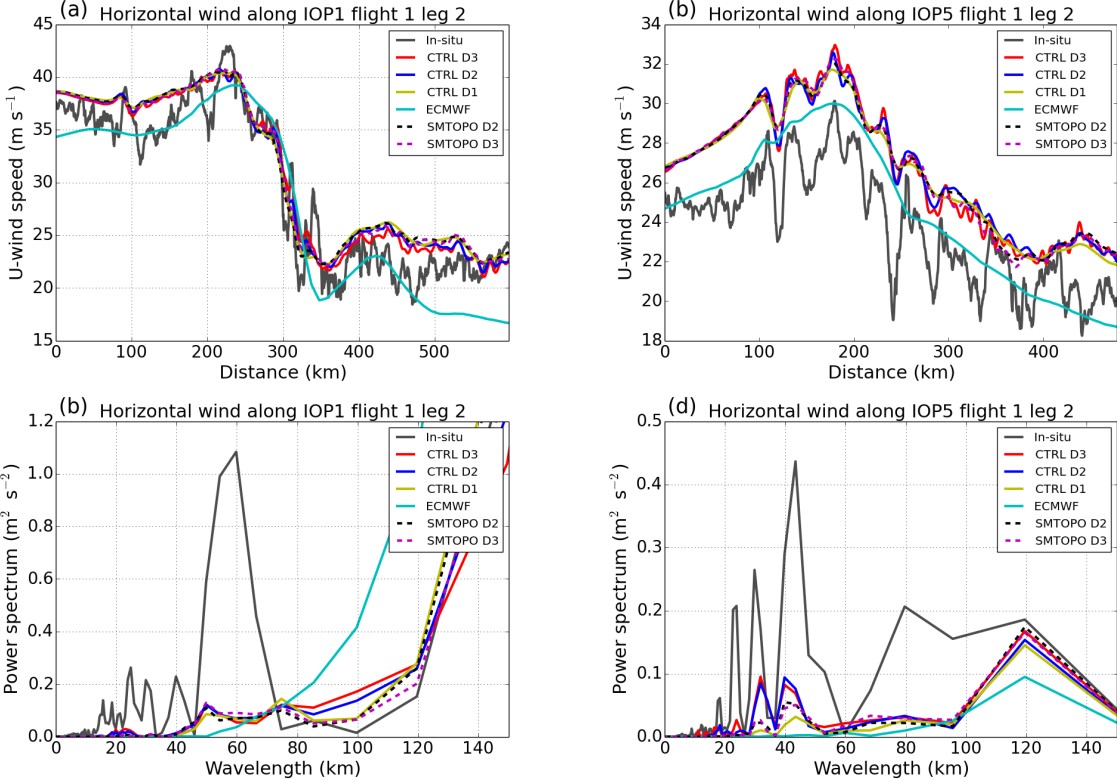

**Figure 13.** As in Fig. 12, but for horizontal wind from a direction of 300° at flight level.

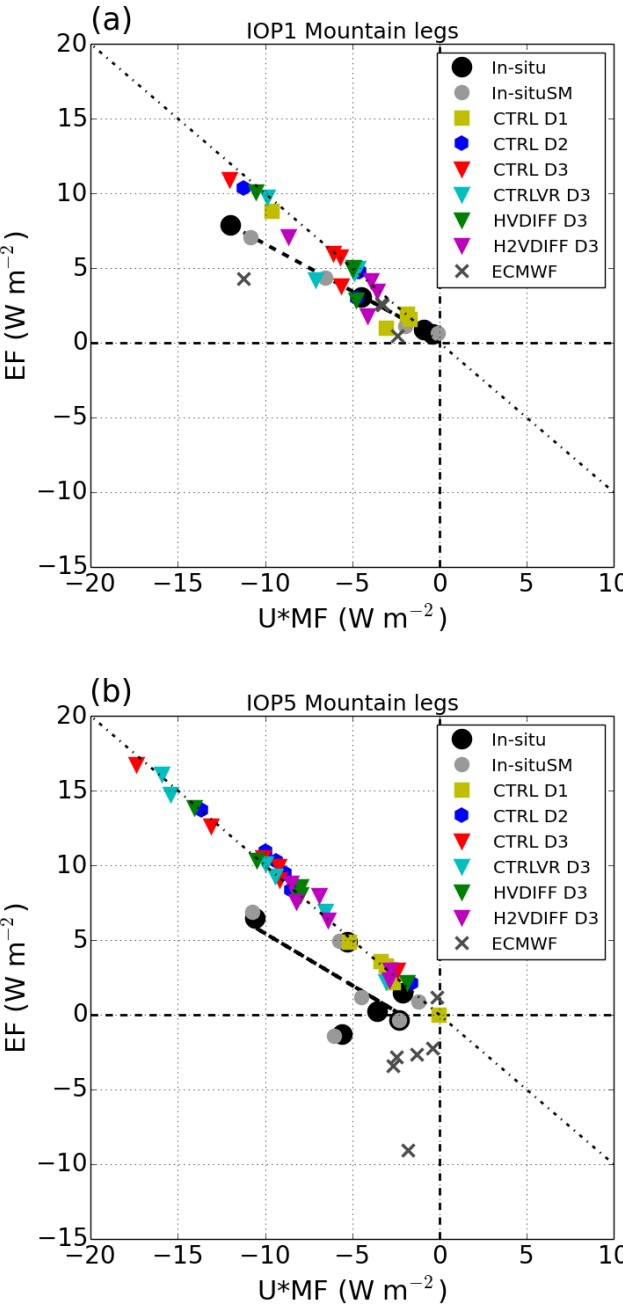

**Figure 14.** Leg-averaged Eliassen-Palm relation between energy flux (EF) and momentum flux (MF) multiplied by leg averaged horizontal wind speed U along all flight legs during (a) IOP1 and (b) IOP5 for observed and simulated data at flight level. The dash dotted line marks the identity line and the thick dashed line indicates the linear regression of observed data. The grey dots (In-situSM) mark observed data for wavelengths larger than 15 km. To exclude effects of non-orographic GWs only data directly over the mountains are utilized (see Fig. 2 for leg locations).

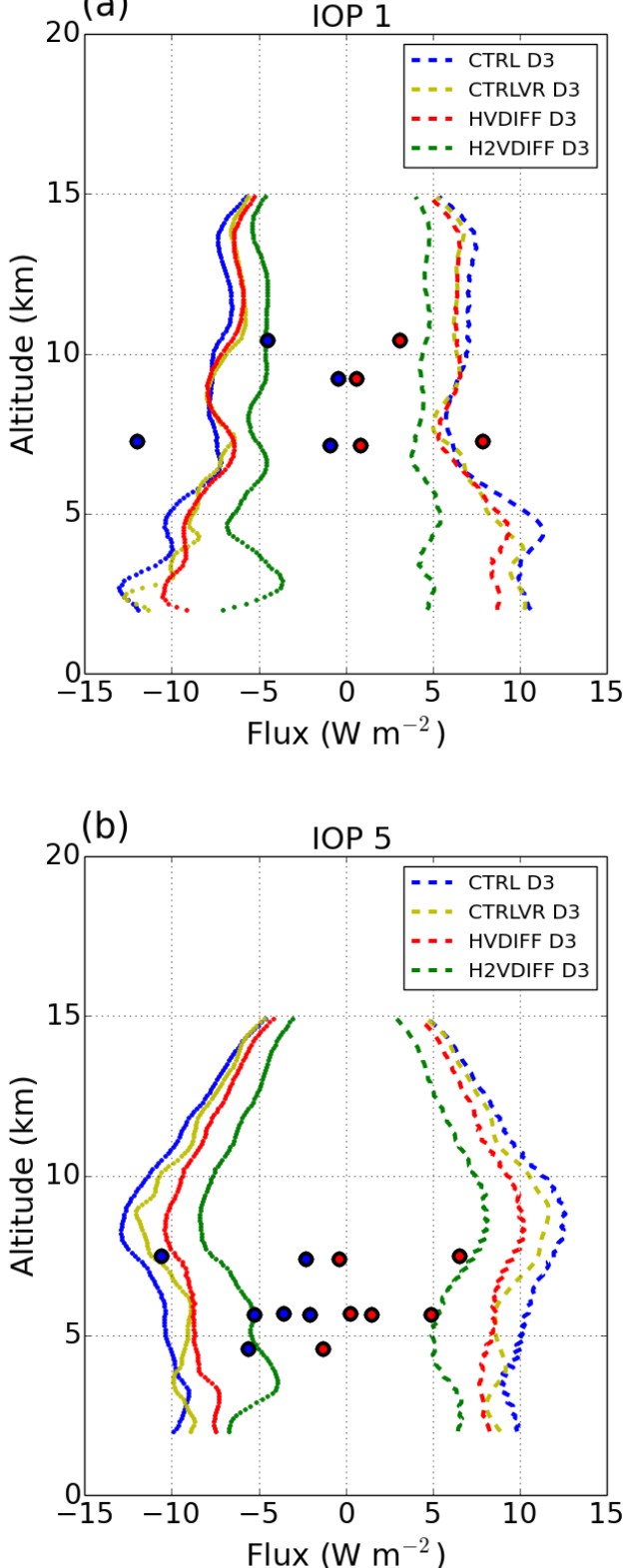

**Figure 15.** Profiles of EF (dashed lines) and MF (dotted lines) averaged over all flight legs during (a) IOP1 and (b) IOP5 for different

sensitivity runs of domain D3. Red and blue dots indicate EF and MF obtained from in-situ measurements of single flight legs (same as in

Fig. 14).

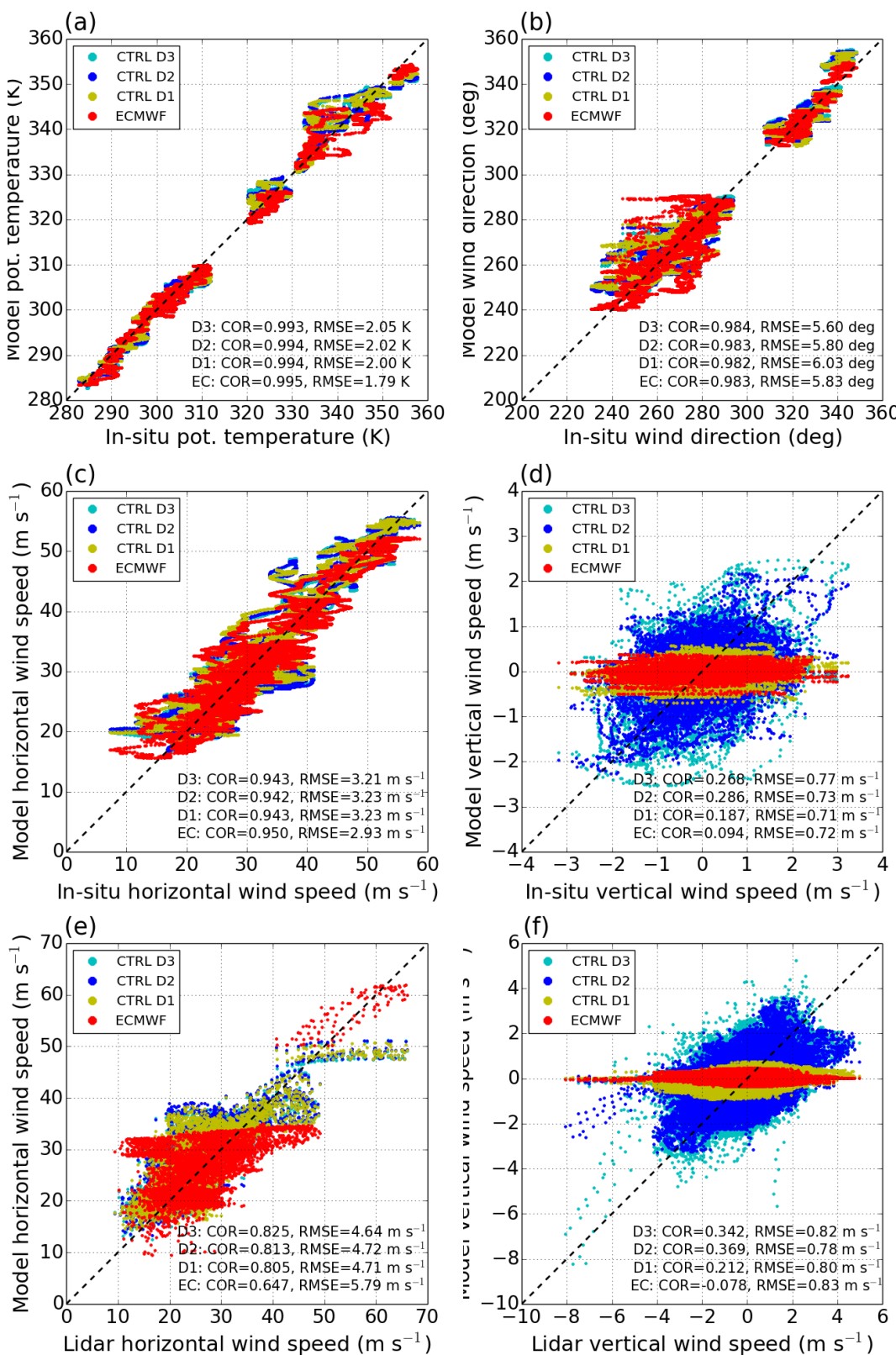

**Figure 16.** Correlation between in-situ and simulated (a) potential temperature, (b) wind direction, (c) horizontal and (d) vertical wind speed at flight level along all flight legs during IOP1 and IOP5. The correlation between simulations and lidar observations is shown in (e) for horizontal and in (f) for vertical wind speed along all lidar cross sections during IOP1 and IOP5. The identity line is marked with the dashed line. Correlation coefficients (COR) and root mean square errors (RMSE) are shown on the bottom of each figure.

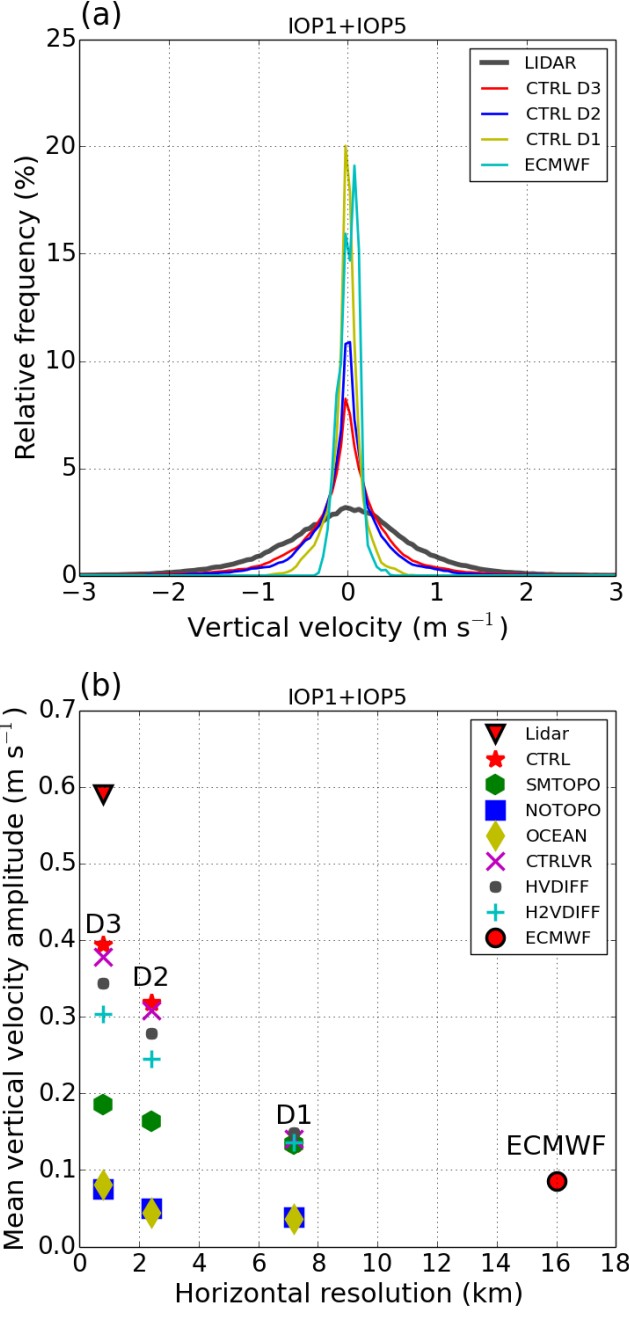

**Figure 17.** Lidar and model vertical velocity distribution (a) and mean vertical velocity amplitude in dependence of horizontal model grid

resolution (b) for all lidar nadir pointing flight legs during both IOP1 and IOP5 (cf. Fig 2).