# Peer review of "Manuscript prepared for Atmos. Chem. Phys."

_Atmospheric Chemistry and Physics, 2016_

## Short Comment (SC1) · 5 Oct 2016

This manuscript is very interesting and thorough. However, I think that one important issue is missing: you should specify what dt you used to correctly handle GW and avoid possible numerical instabilities, particularly if GW produce large vertical velocities. Whatever dt values you used, sensitivity tests should be performed to test the robustness of your results and verify if some dt intervals lead to better agreement with observations.

---

## Referee Comment (RC1) · Anonymous Referee #1 · 25 Oct 2016

The manuscript presents a case study of mountain waves observed over Scandinavia during a field campaign that took place in the December 2013. Two orographic wave events were analyzed using field observations and numerical simulations. These two events were simulated using global and mesoscale models over a range of resolutions and with real, smoothed, or no terrain to test the sensitivity of the simulated waves to model resolutions and resolved topography. The simulated waves and wave energy and momentum fluxes were compared with field observations. Their simulations with higher resolutions reproduce some gross features of the waves qualitatively similar to the lidar and airborne in-situ measurements. The authors showed that it is necessary to have high model resolutions and better resolved topography to simulate the observed

trapped waves, which usually have shorter wave length than propagating hydrostatic waves. In my opinion, their diagnosis methods in general are sound and the results look reasonable. The manuscript is well written, more or less, and the overall figure quality is pretty good. There are a couple of issues that bother me. First, the authors put a lot of emphasis on trapped waves and tropopause reflection. However, I don't think they actually demonstrated that the waves they referred to were trapped waves, or the atmospheric conditions supported trapped waves. Secondly, overall, this manuscript reads more like a technical report instead of a scientific paper. This can be seen from the Conclusion section, which mostly just recaps what's been done. The only conclusion from this study seems to be that topography needs to be well resolved in order to simulate short gravity waves. Of course, this is interesting, but not new at all. It has been known for decades and is the reason for gravity wave drag parameterization in coarse global models. I think there is still plenty of room for improvement before being accepted for publication, and some suggestions are listed below. 1) For the trapped wave case, the authors need to show that those are actually trapping waves, beyond speculation. The vertical cross-section plots and w fluctuations along flight legs are too noisy to tell which and where are trapped waves. The authors showed Scorer parameter profiles calculated from their control simulations, which is helpful, and yet they didn't discuss much about the implication of these profiles. For example, from Fig. 8, it seems that only waves shorter than 30 km may be trapped bellow 5 km. However, in the abstract, the trapped waves ranged from 15 to 40 km. There are a few things they can do to support their argument: a. Solve linear wave equations (e.g., Taylor-Goldstein) for trapped wave modes using observed and simulated profiles, and hope that the observed and simulated trapped waves are consistent with linear wave solutions. b. Redo their idealized solutions using profiles approximated from the real profiles and hope the idealized solutions produce trapped waves with wavelengths comparable to the observations. c. Check phase relations between different variables and hope they are consistent with trapped waves. 2) The role tropopause plays in wave reflection was repeatedly mentioned in the text to explain wave trapping, negative

energy flux, etc. I don't quite follow the argument. Firstly, it seems that waves were trapped in the lower troposphere and, if so, why the tropopause reflection played a role in wave trapping (line 20, abstract)? Secondly, GW can be reflected by sharp change in stratification or wind, or by wave breaking zone. How can the authors tell it was the tropopause that did the reflection? Again, there are a few things they can do and should do here: a. Figure out where and by what the waves were trapped. If the waves were trapped between the tropopause and the ground surface. b. Repeat the simulation with higher vertical resolution near the tropopause to see if the reflected fluxes increase and the up-going fluxes decrease due to the increased resolution, as they speculated (line 22 in abstract and places in text). This could be one of their most important conclusions from this research and shouldn't be built on speculation. c. Compute fluxes at levels right below the wave breaking layer and right below the tropopause to see how much negative energy fluxes at each level. If the latter far exceeded the former, then the authors can conclude, with some confidence, that the tropopause reflection dominates. 3) By the same token, the authors argued that the simulated trapped waves decayed faster than observed because of weakened reflection associated with lower stratification in the tropopause due to low vertical model resolution. Again, we shouldn't make conclusions based on speculation. There are a couple of things that can be done to help make their case. a. As in 2), according to their argument, the trapped waves should decay much slower in their new simulation with high resolution across the tropopause. b. As shown in Smith et al. (2002) and Hills et al. (2016), there are a number of processes that could dissipate trapped waves and caused the rapid decay of their amplitudes with downwind distance. The authors could test the relative importance in their idealized framework.

---

## Referee Comment (RC2) · Anonymous Referee #2 · 30 Oct 2016

General comments

The authors investigate two mountain wave events over Scandinavia within the GWL-CYCLE campaign using measurements and simulations. The campaign has a lot of valuable measurements such as airborne in-situ and lidar observations, which allow the authors to analyse the gravity wave (GW) observations in the upper troposphere and compare them with the simulated ones. The presentation of the observations is relevant by itself, showing the gravity wave events. On the other hand, simulations are focussed on exploring the horizontal resolution sensitivity and the topography influence when resolving the GWs, which are very interesting tests. Results show that topography needs to be resolved in order to capture the proper GWs and an horizontal grid

around 2.4 km seems to be enough. Simulated GWs also seem to reproduce too small amplitudes and too much decaying, which is a significant result. The momentum and energy fluxes are also investigated, showing that simulations tend to overestimate them comparing with observations. The text is well written and figures and tables are clearly exposed, helping to understand and to illustrate the main results. However, there are a few comments I would like to point out:

1. Have you tested other vertical level resolution? Does the vertical levels affect the resolution of mountain waves? Maybe near the stratosphere the number of levels is important and changes the resolved inversion layer. Could you run a test increasing the vertical resolution near the tropopause?

2. The main conclusion is that simulations with mesh sizes larger than 2.4 km cannot simulate the tropospheric GWs. What can you say about the differences between 0.8 km and 2.4 km? Have you checked if the waves propagate within the boundary layer? Are there any differences between these two simulations?

3. There could be more references in section 5 comparing the obtained results with other previous works.

Minor comments

Lines 257-262: It seems that during IOP1 there is a GW breaking at altitudes between 25 and 30 km and the horizontal wind speeds are reduced at these levels at the same time. Does it mean that when GW break the wind speed is reduced? Please explain or provide a reference.

Line 269: "due to critical level dissipation". It is the first time in the manuscript that appears the concept of the "critical level". Could you add a sentence explaining better what is the critical level? Or maybe you could introduce that concept earlier in the introduction.

Line 305: I would say the GWs have weaker amplitudes, not the vertical wind. In

addition, why do you think vertical winds are weaker compared to observations?

Line 382: "the change in stability at the tropopause is more distinct". Do you mean the inversion is much stronger at the tropopause level? Could you reformulate the sentence to be easier to understand?

Figure 3. Colors of CTRL D2 (blue) and CTRL D3 (green) can be easily confused, and lines cannot be distinguished. I would recommend using different colors.

Page 28, caption of Figure 4. "..black and grey lines..". I guess black line for IOP1 is the green line in the plots, so "black" should be replaced by "green".

---

## Author Comment (AC2) · 8 Feb 2017

The comment was uploaded in the form of a supplement:
http://www.atmos-chem-phys-discuss.net/acp-2016-765/acp-2016-765-AC2-supplement.pdf

---

## Author Response (AR1)

**Observed versus simulated mountain waves over Scandinavia - improvement by enhanced model resolution?**

**Reply to interavtive comments (acp-2016-765-SC1) of manuscript acp-2016-765**

Johannes Wagner et al.

February 7, 2017

**1 Introduction**

We thank Peter Alexander for the interactive comment and acknowledge his effort to improve our manuscript.

In the following, interactive comments are marked with numbers and corresponding replies of the authors are written in bold and labelled with " $\Rightarrow$ ".

- 1. This manuscript is very interesting and thorough. However, I think that one important issue is missing: you should specify what dt you used to correctly handle GWs and avoid possible numerical instabilities, particularly if GWs produce large vertical velocities.
- ⇒ We agree that information about the used time step is missing in the manuscript and added the following sentence in section 3.1 (P7, L149): To avoid numerical instabilities adaptive time stepping was used with a maximum time step of 15 s and a maximum Courant number of 1.2.
- 2. Whatever dt values you used, sensitivity tests should be performed to test the robustness of your results and verify if some dt intervals lead to better agreement with observations.
- ⇒ We think that testing the impact of different time steps on model results would be quite interesting. As this study focuses, however, on different spatial resolutions of the model grid, additional simulations with varying time steps would go beyond the scope of this study.

**Observed versus simulated mountain waves over Scandinavia - improvement by enhanced model resolution?**

**Reply to comments of anonymous referee 1 of manuscript acp-2016-765**

Johannes Wagner et al.

February 7, 2017

**1 Introduction**

We thank the anonymous referee for the comments and acknowledge his effort to improve our manuscript. We performed three additional real-case sensitivity simulations and 2D idealized simulations along the flight legs to clarify the meteorological situation and to improve vertical energy and momentum fluxes. The previous idealised simulations (HYDRO and TRAPPED) were removed as their set-up was more or less arbitrarily chosen and did not have a direct reference to the case study. We also removed the comparison of the observed and simulated reflection coefficient at the tropopause (former Fig. 15), as further investigations showed, that wave trapping did not occur at the tropopause. In addition, we revised the introduction and changed the title to "Observed versus simulated mountain waves over Scandinavia - improvement of vertical winds, energy and momentum fluxes by enhanced model resolution?" to emphasize the focus on GW-induced vertical winds and energy and momentum fluxes.

In the following, comments of the referee are marked with numbers and corresponding replies of the authors are written in bold and labeled with " $\Rightarrow$ ".

**2 Summary**

The manuscript presents a case study of mountain waves observed over Scandinavia during a field campaign that took place in December 2013. Two orographic wave events were analyzed using field observations and numerical simulations. These two events were simulated using

global and mesoscale models over a range of resolutions and with real, smoothed, or no terrain to test the sensitivity of the simulated waves to model resolutions and resolved topography. The simulated waves and wave energy and momentum fluxes were compared with field observations. Their simulations with higher resolutions reproduce some gross features of the waves qualitatively similar to the lidar and airborne in-situ measurements. The authors showed that it is necessary to have high model resolutions and better resolved topography to simulate the observed trapped waves, which usually have shorter wave lengths than propagating hydrostatic waves. In my opinion, their diagnosis methods in general are sound and the results look reasonable. The manuscript is well written, more or less, and the overall figure quality is pretty good. There are a couple of issues that bother me.

**3 Comments**

- 1. First, the authors put a lot of emphasis on trapped waves and tropopause reflection. However, I don't think they actually demonstrated that the waves they referred to were trapped waves, or the atmospheric conditions supported trapped waves.
- ⇒ We agree and performed additional 2D idealised simulations along the flight legs to simplify the meteorological situation and to demonstrate the formation of interfacial waves during IOP5 along a stratospheric intrusion at an altitude of about 5 km. It is true that interfacial waves were very weak during IOP1 and did not occur in 2D idealised simulations of this event (see section 3.2 and 4.3). Interfacial waves in idealized simulations of IOP5 had the same horizontal wavelength of about 10 km as interfacial waves in CTRL simulations (see Fig. 6, Fig. 10 and section 4.3).
- 2. Secondly, overall, this manuscript reads more like a technical report instead of a scientific paper. This can be seen from the conclusion section, which mostly just recaps what's been done. The only conclusion from this study seems to be that topography needs to be well resolved in order to simulate short gravity waves. Of course, this is interesting, but not new at all. It has been known for decades and is the reason for gravity wave drag parameterization in coarse global models.
- ⇒ We agree that especially the last sections of the paper repeated already discussed issues. Thus, we revised the conclusion section. By including the additional sensitivity runs we particularly demonstrate that our paper shows the skills and problems of a state of the art mesoscale model when simulating gravity waves. We show that even with horizontal resolutions of 800 m the observed wave field cannot be captured completely and that there are relatively large disagreements between observed and simulated energy and momentum fluxes.

3. I think there is still plenty of room for improvement before being accepted for publication, and some suggestions are listed below. 1) For the trapped wave case, the authors need to show that those are actually trapping waves, beyond speculation. The vertical cross-section plots and w fluctuations along flight legs are too noisy to tell which and where are trapped waves. The authors showed Scorer parameter profiles calculated from their control simulations, which is helpful, and yet they didn't discuss much about the implication of these profiles. For example, from Fig. 8, it seems that only waves shorter than 30 km may be trapped bellow 5 km. However, in the abstract, the trapped waves ranged from 15 to 40 km. There are a few things they can do to support their argument: a. Solve linear wave equations (e.g., Taylor-Goldstein) for trapped wave modes using observed and simulated profiles, and hope that the observed and simulated trapped waves are consistent with linear wave solutions.

b. Redo their idealized solutions using profiles approximated from the real profiles and hope the idealized solutions produce trapped waves with wavelengths comparable to the observations.

c. Check phase relations between different variables and hope they are consistent with trapped waves.

- ⇒ We agree that additional effort had to be done to clarify the meteorological situation. We used your proposal b) and performed idealised 2D simulations along the two example flight legs during both IOP1 and IOP5 (see Fig. 9 and section 4.3). Idealised simulations were initialised with mean upstream profiles of CTRL D3 simulations (Fig. 8) and show the wave formation under simplified conditions. It becomes clear that during IOP1 no (or very weak) interfacial waves occured along a tropopaus fold. During IOP5 waves were ducted in a stratified layer around 5 km altitude.
- 4. 2) The role tropopause plays in wave reflection was repeatedly mentioned in the text to explain wave trapping, negative energy flux, etc. I don't quite follow the argument. Firstly, it seems that waves were trapped in the lower troposphere and, if so, why the tropopause reflection played a role in wave trapping (line 20, abstract)?
- ⇒ We acknowledge the comment and adapted statements in the text to the new results obtained from idealised simulations. It is true that wave trapping did not occur at the tropopause during IOP5, but at a stratospheric intrusion layer at lower altitudes.
- 5. Secondly, GW can be reflected by sharp change in stratification or wind, or by wave breaking zone. How can the authors tell it was the tropopause that did the reflection? Again, there are a few things they can do and should do here: a. Figure out where and by what the waves were trapped. If the waves were trapped between the tropopause and the ground surface.

**⇒ We agree and found that wave trapping occured at a stratified layer at about 5 km altitude during IOP5.**

- 6. b. Repeat the simulation with higher vertical resolution near the tropopause to see if the reflected fluxes increase and the up-going fluxes decrease due to the increased resolution, as they speculated (line 22 in abstract and places in text). This could be one of their most important conclusions from this research and shouldn't be built on speculation.
- ⇒ We performed an additional sensitivity run with increased vertical grid resolution, which has a constant level distance of 80 m throughout the troposphere and lower stratosphere (CTRLVR, see Table 2). These simulations improved the leg-averaged energy and momentum fluxes by up to 2 W m-2 (Fig. 15). Additionally we performed two sensitivity runs with increased turbulent diffusion (HVDIFF and H2VDIFF, see Table 2 and section 3.1) to enhance nonlinear wave effects. These simulations showed significantly reduced energy and momentum fluxes especially for the H2VDIFF run (Fig. 14 and Fig. 15).
- 7. c. Compute fluxes at levels right below the wave breaking layer and right below the tropopause to see how much negative energy fluxes at each level. If the latter far exceeded the former, then the authors can conclude, with some confidence, that the tropopause reflection dominates.
- $\Rightarrow$  We computed flux profiles of all D3 simulations (Fig. 15) to show that fluxes were reduced in simulations with increased turbulent diffusion (section 5.2).
- 8. 3) By the same token, the authors argued that the simulated trapped waves decayed faster than observed because of weakened reflection associated with lower stratification in the tropopause due to low vertical model resolution. Again, we shouldn't make conclusions based on speculation. There are a couple of things that can be done to help make their case. a. As in 2), according to their argument, the trapped waves should decay much slower in their new simulation with high resolution across the tropopause.
- ⇒ We performed additional simulations with higher vertical grid resolution (CTR-LVR), which did slightly improve the energy and momentum fluxes but not the decay of waves in the lee of the mountains (not shown). We therefore left out the discussion about lee wave decay.
- 9. b. As shown in Smith et al. (2002) and Hills et al. (2016), there are a number of processes that could dissipate trapped waves and caused the rapid decay of their amplitudes with downwind distance. The authors could test the relative importance in their idealized framework.
- ⇒ We think that it would be interesting to investigate reasons for stronger decay of trapped waves by means of the idealised simulations. This would, however, go beyond the scope of this study and we focused on the improvement of energy and momentum fluxes by means of additional real-case simulations.

**Observed versus simulated mountain waves over Scandinavia - improvement by enhanced model resolution?**

**Reply to comments of anonymous referee 2 of manuscript acp-2016-765**

Johannes Wagner et al.

February 7, 2017

**1 Introduction**

We thank the anonymous referee for the comments and acknowledge his effort to improve our manuscript. We performed three additional real-case sensitivity simulations and 2D idealized simulations along the flight legs to clarify the meteorological situation and to improve vertical energy and momentum fluxes. The previous idealised simulations (HYDRO and TRAPPED) were removed as their set-up was more or less arbitrarily chosen and did not have a direct reference to the case study. We also removed the comparison of the observed and simulated reflection coefficient at the tropopause (former Fig. 15), as further investigations showed, that wave trapping did not occur at the tropopause. In addition, we revised the introduction and changed the title to "Observed versus simulated mountain waves over Scandinavia - improvement of vertical winds, energy and momentum fluxes by enhanced model resolution?" to emphasize the focus on GW-induced vertical winds and energy and momentum fluxes.

In the following, comments of the referee are marked with numbers and corresponding replies of the authors are written in bold and labeled with " $\Rightarrow$ ".

**2 General comments**

The authors investigate two mountain wave events over Scandinavia within the GWLCYCLE campaign using measurements and simulations. The campaign has a lot of valuable measurements such as airborne in-situ and lidar observations, which allow the authors to analyse the

gravity wave (GW) observations in the upper troposphere and compare them with the simulated ones. The presentation of the observations is relevant by itself, showing the gravity wave events. On the other hand, simulations are focussed on exploring the horizontal resolution sensitivity and the topography influence when resolving the GWs, which are very interesting tests. Results show that topography needs to be resolved in order to capture the proper GWs and an horizontal grid around 2.4 km seems to be enough. Simulated GWs also seem to reproduce too small amplitudes and too much decaying, which is a significant result. The momentum and energy fluxes are also investigated, showing that simulations tend to overestimate them comparing with observations. The text is well written and figures and tables are clearly exposed, helping to understand and to illustrate the main results. However, there are a few comments I would like to point out:

**3 Major comments**

- 1. Have you tested other vertical level resolution? Does the vertical levels affect the resolution of mountain waves? Maybe near the stratosphere the number of levels is important and changes the resolved inversion layer. Could you run a test increasing the vertical resolution near the tropopause?
- ⇒ We acknowledge the comment and performed an additional sensitivity run with increased vertical grid resolution, which has constant level distances of 80 m in the troposphere and lower stratosphere (CTRLVR, see Table 2). Leg-averaged energy and momentum fluxes of these simulations were slightly reduced by up to 2 W m-2 (Fig. 15, section 5.2). In addition we performed two further sensitivity runs with increased turbulent diffusion (HVDIFF and H2VDIFF, see section 3.1) to amplify non-linear wave effects. Energy and momentum fluxes of the H2VDIFF simulation were significantly decreased up to 6 W m-2 compared to the CTRL run fields (see Fig. 14 and Fig. 15).
- 2. The main conclusion is that simulations with mesh sizes larger than 2.4 km cannot simulate the tropospheric GWs. What can you say about the differences between 0.8 km and 2.4 km? Have you checked if the waves propagate within the boundary layer? Are there any differences between these two simulations?
- ⇒ The simulations show that the wave patterns are captured well in the D2 runs. However, on average vertical wind speeds are nearly 0.1 m s-1 smaller than in D3 runs and especially small-scale interfacial waves are weaker in the D2 simulations. Cross sections of both real-case and idealised simulations show, that waves also propagate in the extremely stable boundary layer (Fig. 6 and Fig. 9). Further idealised simulations with different boundary layer characteristics and their impact on the GW evolution were, however, not performed in this study.

- 3. There could be more references in section 5 comparing the obtained results with other previous works.
- $\Rightarrow$  We agree and included some more references.

**4 Minor comments**

- 4. Lines 257-262: It seems that during IOP1 there is a GW breaking at altitudes between 25 and 30 km and the horizontal wind speeds are reduced at these levels at the same time. Does it mean that when GW break the wind speed is reduced? Please explain or provide a reference.
- $\Rightarrow$  We added a reference in the text (L265).
- 5. Line 269: "due to critical level dissipation". It is the first time in the manuscript that appears the concept of the "critical level". Could you add a sentence explaining better what is the critical level? Or maybe you could introduce that concept earlier in the introduction.
- ⇒ We added the following explanation in the text (L275): "This means that the growing wave amplitude generates regions with nearly zero winds while the vertical wavelength approaches zero. This leads to convective overturning and turbulent wave breaking (?)"
- 6. Line 305: I would say the GWs have weaker amplitudes, not the vertical wind. In addition, why do you think vertical winds are weaker compared to observations?
- $\Rightarrow$  We agree and changed the formulation in the text (L339). We think that amplitudes are weaker mainly due to numeric diffusion and added this in the text (L340).
- 7. Line 382: "the change in stability at the tropopause is more distinct". Do you mean the inversion is much stronger at the tropopause level? Could you reformulate the sentence to be easier to understand?
- $\Rightarrow$  We left out this sentence and the comparison between observed and simulated stratification at the tropopause, as wave trapping occured along a stratified layer at an altitude of about 5 km.
- 8. Figure 3. Colors of CTRL D2 (blue) and CTRL D3 (green) can be easily confused, and lines cannot be distinguished. I would recommend using different colors.
- $\Rightarrow$  We agree and changed the colors in Figs. 3, 7, 8, 12, 13 and 17.
- 9. Page 28, caption of Figure 4. "...black and grey lines...". I guess black line for IOP1 is the green line in the plots, so "black" should be replaced by "green".

 $\Rightarrow$  The complete flights are marked with grey and black lines. Only the example flight leg 2 of the respective first research flights during both IOP1 and IOP5, which is used in this study is marked with a green line. We added this in the figure caption (Fig. 4).

Manuscript prepared for Atmos. Chem. Phys.

with version 2014/05/15 6.81 Copernicus papers of the LATEX class copernicus.cls.

Date: 8 February 2017

**Observed versus simulated mountain waves over Scandinavia improvement of vertical winds, energy and momentum fluxes by enhanced model resolution?**

Johannes Wagner1, Andreas Dörnbrack1, Markus Rapp1, Sonja Gisinger1, Benedikt Ehard1, Martina

5 Bramberger1, Benjamin Witschas1, Fernando Chouza1, Stephan Rahm1, Christian Mallaun2, Gerd Baumgarten3, and Peter Hoor4

[revised manuscript text omitted]
   | 0.992 0.998 | 0.891 0.967 | 0.116 0.076 | 0.858 0.897 | 0.414 0.882 | -0.064 -0.092 |